# MATRYOSHKA: LEARNING TO DRIVE BLACK-BOX LLMS WITH LLMS

## ABSTRACT

Despite the impressive generative abilities of black-box large language models (LLMs), their inherent opacity hinders further advancements in capabilities such as reasoning, planning, and personalization. Existing works aim to enhance LLM capabilities via domain-specific adaptation or in-context learning, which require additional training on accessible model parameters, an infeasible option for black-box LLMs. To address this challenge, we introduce Matryoshka, a lightweight white-box LLM controller that guides a large-scale black-box LLM generator by decomposing complex tasks into a series of intermediate outputs. Specifically, we consider the black-box LLM as an environment, with Matryoshka serving as a policy to provide intermediate guidance through prompts for driving the black-box LLM. Matryoshka is trained to pivot the outputs of the black-box LLM aligning with preferences during iterative interaction, which enables controllable multi-turn generation and self-improvement in optimizing intermediate guidance. Empirical evaluations on three diverse tasks demonstrate that Matryoshka effectively enhances the capabilities of black-box LLMs in complex, long-horizon tasks, including reasoning, planning, and personalization. By leveraging this pioneering controller-generator framework to mitigate dependence on model parameters, Matryoshka provides a transparent and practical solution for improving black-box LLMs through controllable multi-turn generation using white-box LLMs.

## 1 INTRODUCTION

Most of the commercial large language models (LLMs) (Radford et al., 2019; Brown, 2020; Achiam et al., 2023; Chowdhery et al., 2023; Team et al., 2023; Reid et al., 2024) are *black-box* models (Sun et al., 2024b; Zhuang et al., 2024), where the model structure, parameters, or even output logits are not accessible. Although these black-box LLMs have exhibited remarkable efficacy across a diverse array of applications, revolutionizing natural language processing tasks such as text completion (Radford et al., 2019; Brown, 2020), translation (Zhu et al., 2023), question-answering (Hendrycks et al., 2020), etc, the applications of black-box LLMs continue to face significant challenges when faced with tasks that require more advanced cognitive capabilities, particularly in the realms of reasoning (Hendrycks et al., 2021; Wang et al., 2024b), planning (Valmeekam et al., 2022; Zhuang et al., 2023; Jimenez et al., 2023; Mialon et al., 2023), and personalization problems (Salemi et al., 2023; Tan et al., 2024a). Enhancing such capabilities within black-box LLMs presents unique challenges, primarily due to the *lack of direct access to internal model parameters* (Huang et al., 2023; Sun et al., 2024b; Zhuang et al., 2024). This opacity introduces substantial complexity in efforts to refine and augment these advanced cognitive functions within the framework of black-box architectures.

Existing research efforts for improving black-box LLM performance can be largely categorized into two main methodological paradigms (Figure 1): (1) **In-context learning (ICL)-based methods** (Sun et al., 2024a; Tan et al., 2024b; Zhuang et al., 2024) that are designed to guide LLM in exhibiting specific capabilities or adhering to particular directives. However, these frameworks necessitate *meticulously constructing few-shot demonstrations or prompts* for LLMs to emulate or follow, rather than fundamentally advancing their intrinsic capabilities. (2) **Adapter-based methods** (Sun et al., 2024b; Zhuang et al., 2024; Shi et al., 2024) that exploit the inherent randomness in LLM generation, producing multiple candidate outputs and subsequently selecting those that optimally satisfy domain-predetermined criteria. Nevertheless, these approaches are highly dependent on the intrinsic synthetic capabilities or built-in functionalities of the black-box LLM, potentially resulting in the selection of a

Figure 1: Enhancement in black-box LLMs capabilities. Existing methods either (a) integrate well-crafted instructions or meticulously-picked few-shot demonstrations as guidance or (b) exploit randomness in model generations to identify the most promising solution from candidates. In `Matryoshka`, we present (c) a controller-generator framework that enables white-box LLMs to drive the behavior of black-box LLMs for enhanced capabilities. 🔥 indicates the trainable parameters, whereas ❄ indicates the inaccessible fixed parameters.

*suboptimal candidate* when all the generated options are less than ideal. Furthermore, both ICL and adapter-based methodologies exhibit significant limitations when applied to long-horizon tasks (*e.g.*, multi-step reasoning, long-term planning, *etc.*) due to their inherent lack of environmental interaction capabilities. In light of these constraints, we propose to leverage smaller, open-source LLMs as controllers to generate soft prompts as guidance, instead of relying on hard memory in context.

Similar to the scratchpad in `o1-preview` (OpenAI, 2024c)[1], we propose `Matryoshka`, a modular framework designed to enhance the advanced problem-solving capabilities of black-box LLMs via controllable multi-turn generations. `Matryoshka` consists of a lightweight white-box LLM that functions as a **controller** and a black-box LLM that serves as a **generator** or **solver**. Upon receiving the question description as input, the controller generates intermediate outputs that augment the capabilities of the subsequent black-box LLMs. For example, the controller can decompose the original complex task into high-level subtasks in reasoning or planning scenarios, or summarize profiles from historical records for personalization tasks. By conceptualizing the following **black-box LLM as the environment**, `Matryoshka` generates intermediate guidance alongside the original input to derive the final result through multi-turn interactions with the environment. The feedback for the outputs from the environments distinguishes positive and negative examples of intermediate generations, which can be used for preference optimization. Notably, this optimization process is inherently self-improving through iterative sampling from prior inferences and by considering the policies from earlier iterations as reference policies. `Matryoshka` continually enhances the advanced capabilities of the black-box LLM through controllable multi-turn generations that iteratively interact with environmental feedback.

Extensive experiments conducted on three complex tasks demonstrate the effectiveness and generalizability of `Matryoshka` in improving the advanced problem-solving capabilities of black-box LLMs, with an average improvement of 3.19% in accuracy for reasoning, 7.46% in success rate for planning, and 5.82% in accuracy for personalization. Importantly, `Matryoshka` not only enhances the capabilities of black-box LLMs without requiring access to model parameters, but also facilitates online feedback with environmental interactions. We summarize the main contributions as follows:

- **i),** We introduce `Matryoshka`, one of the first modular frameworks that employ a lightweight white-box LLM to drive the generation of a large-scale black-box LLM for complex problem-solving;
- **ii),** `Matryoshka` intuitively formulates the white-box LLM as a controller and the black-box LLM as a component of the environment, facilitating long-horizon controllable generation with environmental feedback; and
- **iii),** `Matryoshka` adopts on-policy learning to iteratively enhance training data quality, inherently self-improving intermediate guidance for the continual enhancement of black-box LLM capabilities.

## 2 PROBLEM FORMULATION

Our objective is to enhance the capability of a black-box LLM in solving complex, long-horizon problems by calibrating its output generation to better align with specific tasks. To achieve this, we conceptualize both the original outputs and the optimal solutions as distributions within a joint space, $\mathcal{Y} \sim \mathcal{Y}^{\text{org}} \times \mathcal{Y}^{\text{sol}}$, where $\mathcal{Y}^{\text{org}}$ and $\mathcal{Y}^{\text{sol}}$ represent the original text generations and target solutions,

---

[1]Scratchpad is a sequence of intermediate chain-of-thoughts generated prior to producing the final answer. In `Matryoshka`, we broaden the definition of intermediate tokens to encompass various forms of guidance that can enhance the capabilities of LLMs, including task decomposition and user history summarization.

Figure 2: Controller-generator framework in `Matryoshka` comprising a white-box LLM as the controller and a black-box LLM as the generator and part of the environment. Given an input query $x$, `Matryoshka` leverages the intermediate generation $f_\theta(x)$ from the controller $\theta$ to drive the generator's behavior. The final answer is derived from the generation $y \sim g(f_\theta(x))$.

respectively. Specifically, given a set of task descriptions $\mathcal{D} = \{x_i\}_{i=1}^N$, our goal is to adjust the outputs $\hat{y}_i \in \mathcal{Y}^{\text{org}}$ of the black-box LLM toward the hidden target solutions $y_i \in \mathcal{Y}^{\text{sol}}$ that successfully solve the problems. This involves driving the black-box LLM to generate outputs more closely aligned with the desired solutions without requiring access to its internal parameters.

**White-Box LLM as Controller/Policy.** We propose utilizing a lightweight white-box language model as a controller to enhance the capabilities of black-box LLMs in solving various tasks. The process begins by feeding a text-grounded task description $x$ from the task space $\mathcal{X}$ into a smaller language model $\theta$, which acts as the controller. This smaller model generates $f_\theta(x)$, an automatic prompt designed to augment the performance of black-box LLMs on the specific task. These prompts can facilitate various functions, such as chain-of-thoughts for reasoning, task decomposition for planning, and user profile summarization from historical records for personalization. The generated intermediate prompt $f_\theta(x)$ is then combined with the original problem description $x$ and input $(x, f_\theta(x))$ into the black-box LLM. The capability enhancement will be measured by evaluating the performance improvements achieved through this controller-generator framework.

We emphasize that unlike works focusing on token-level LLM policy (Wang et al., 2024a; Rafailov et al., 2024a), our action space consists of entire intermediate guidance generations to solve the task.

**Black-Box LLM as Environment.** We recognize the black-box LLM as an environment to be controlled by the white-box LLM policy. After inputing the prompts $(x, f_\theta(x))$, the black-box LLM produces a final solution $\hat{y} = g_{\text{LLM}}(x, f_\theta(x))$ for the task. We utilize the final correctness of the black-box LLM's output to evaluate the quality $u(x, f_\theta(x))$ as the reward of the intermediate guidance produced by the white-box LLM controller (Figure 2):

$$u(x, f_\theta(x)) := \text{eval}(x, g_{\text{LLM}}(x, f_\theta(x)), \tag{1}$$

where $\text{eval}(\cdot)$ denotes the oracle evaluation function of the final answer. For example, in question-answering tasks with ground-truth final answer $y$, the evaluation function measures accuracy by comparing the prediction with ground truth as $\text{eval}(x, g_{\text{LLM}}(x, f_\theta(x)) = \mathbb{1}(g_{\text{LLM}}(x, f_\theta(x)) = y)$, where $\mathbb{1}(\cdot)$ is the indicator function. For planning tasks without a ground-truth solution, the evaluation function assesses the success rate after executing the final solution as $\text{eval}(x, g_{\text{LLM}}(x, f_\theta(x)) = \mathbb{1}_{\text{succ}}(g_{\text{LLM}}(x, f_\theta(x)))$.

**Multi-Turn Interaction.** The above interaction between the white-box LLM controller and the black-box environment can be repeated for multi-turns for long-horizon tasks.

For initialization, a prompt $x$ is sampled from task space $\mathcal{X}$ and serves as the initial state $s_0 = x$. At each subsequent step $t \in [T]$, the controller generates prompts $a_t$ based on the current $s_{t-1}$. In response to the controller's action, the environment returns an observation $o_t$ based on the history $s_{t-1}$ and the current action $a_t$. The state then transitions to include the new action and observation:

$$s_t = (s_{t-1}, a_t, o_t) = (x, a_1, o_1, s_1, \cdots, a_t, o_t), \tag{2}$$

and the next step begins. This process repeats for $T$ rounds, resulting in a trajectory:

$$\tau = (x, a_1, o_1, s_1, \cdots, o_T, s_T), \tag{3}$$

and we obtain the reward for the whole trajectories, according to some $\text{eval}(\cdot)$.

The framework formulates a Markov Decision Process (MDP), which offers the potential for solving tasks that require long-horizon generations, including long-term planning and multi-step reasoning. By obtaining feedback from $\text{eval}(\cdot)$, we can conduct multi-turn optimization over the white-box LLM controller on the intermediate generations. Additionally, the multi-turn interaction with the environment during the data sampling stage can help improve data quality. Although optimizing this guidance presents challenges due to the inaccessibility of the black-box LLM's parameters that preclude backpropagation of gradients during training, the existing reinforcement learning techniques, *e.g.*, Schulman et al. (2017); Rosset et al. (2024), can be used for policy optimization.

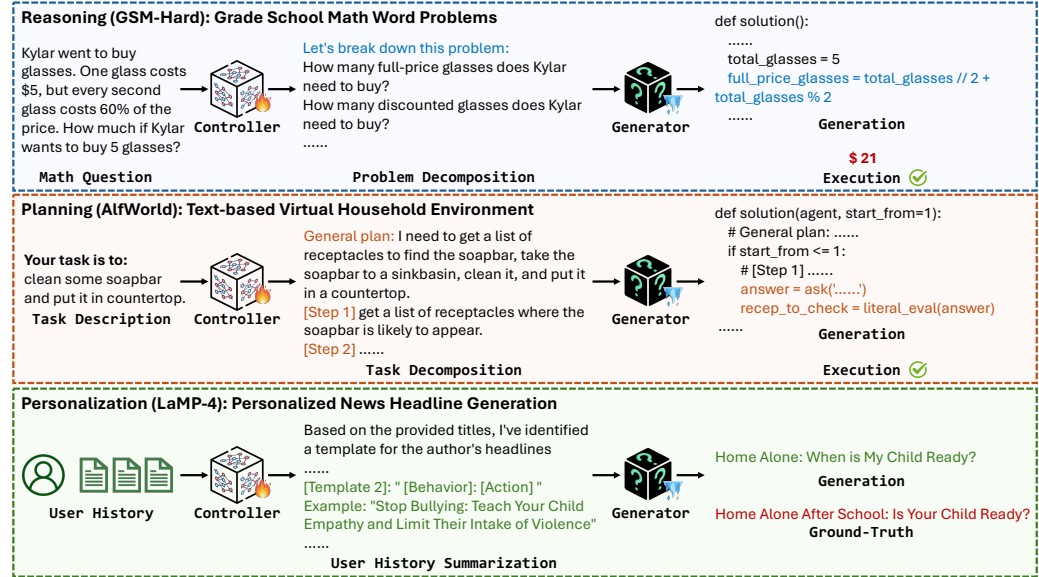

Figure 3: Examples of intermediate guidance generated by `Matryoshka` for complex reasoning, planning, and personalization tasks.

## 3  MATRYOSHKA

In this section, we specialize the white-box LLM controller that generates intermediate guidance to assist in task understanding and problem-solving in Section 3.1 and discuss the data collection procedure by interacting with black-box LLM in Section 3.2, which will be used for `Matryoshka` training to align the outputs of the black-box LLM with preferences in Section 3.3.

### 3.1  INSTANTIATION OF WHITE-BOX LLM CONTROLLER

We instantiate the white-box LLM as a controller to generate additional guidance that assists the black-box LLM in understanding and solving a diverse range of problems. Given the varying complexity and distinct characteristics of different tasks, the controller should be capable of generating guidance in various formats. We provide examples corresponding to reasoning, planning, and personalization tasks (Figure 3):

**Problem Decomposition for Reasoning.** For reasoning tasks, generating a sequence of reasoning steps is essential to solve the problem effectively. Existing works (Zhou et al., 2023) have observed that models often perform poorly on tasks that require solving problems more complex than the exemplars provided in the prompts. To enable the model to develop better reasoning and overcome the easy-to-hard generalization issue, one strategy is to decompose complex problems into a series of simpler sub-problems and solve them sequentially. Therefore, for reasoning tasks, the white-box LLM controller outputs decomposed sub-tasks to assist the subsequent black-box LLM generator in enhancing its reasoning capabilities.

**High-Level Plan for Planning.** For planning tasks, LLMs are required to generate a sequence of actions that constitute a plan to solve the given problems. A common strategy (Sun et al., 2024a; Zhao et al., 2024) is to apply hierarchical planning for complex solutions, where a high-level planner decomposes the task into sub-goals, and a low-level planner generates a sequence of admissible actions corresponding to each specific sub-goal. To enhance the black-box LLM's planning capabilities, we leverage the white-box controller to generate high-level plans as guidance to simplify the problems.

**User History Summarization for Personalization.** For personalization tasks, LLMs are required to tailor outputs to individual users. Existing work (Richardson et al., 2023) accomplishes this by concatenating the user's input query with a profile summarizing the user's preferences and behavior patterns. To enhance the black-box LLM's personalization capabilities, we utilize the white-box LLM controller to generate summaries of user histories. This approach enables black-box LLMs to better understand users and generate tailored content accordingly.

Figure 4: Overview of iterative guidance optimization. By iteratively updating both the model and the reference policy, `Matryoshka` progressively refines its intermediate guidance.

## 3.2 DATA COLLECTION BY INTERACTING WITH BLACK-BOX LLM ENVIRONMENT

Optimizing the intermediate guidance generated by the controller presents significant challenges for two main reasons: (1) *Lack of ground-truth guidance*: There are no ground-truth intermediate generations available to serve as supervision signals for the controller's outputs. (2) *Uncertainty in performance improvement*: It is difficult to determine which guidance will reliably enhance the downstream performance of the black-box LLM. To address these challenges, we formulate the black-box LLM as an environment system and employ multi-turn interactions with environmental feedback during data sampling.

In the MDP formulation, we consider the *action space* as the set of possible guidance that can enhance the capabilities of black-box LLMs. The *observation space* is determined by the oracle evaluation function for each task, defined as $\text{eval}(\cdot)$, where the sampled supervision signal is denoted as $z$, with $z = 1$ indicating that $f_\theta(x)$ is positive guidance while $z = 0$ indicating $f_\theta(x)$ negative guidance. During the multi-turn interactions, if the observation $o_t$ at the $t$-th step returns a negative signal, the next action step $a_{t+1}$ involves modifying the intermediate guidance based on the feedback. The interactions continue until a positive signal is observed or the maximum number of interaction turns $T$ is reached.

For each input $x_i$, we perform $T$-step multi-turn interactions with the black-box LLM-based environment to obtain the trajectories $(a_{i,1}, o_{i,1}, a_{i,2}, o_{i,2}, \cdots, a_{i,T}, o_{i,T})$. To increase the diversity of intermediate generations, we introduce randomness into the policy and repeat the entire interaction process $K$ times. This results in $K$ trajectories, yielding intermediate generations $\{g_{i,1}, g_{i,2}, \cdots, g_{i,K\times T}\}$ along with their corresponding observations $\{o_{i,1}, o_{i,2}, \cdots, o_{i,K\times T}\}$, which serve as sampling signals. We then sample the positive guidance $g_i^+$ from the set of guidance with positive observations, $g_i^+ \sim \{g_{i,j}|o_{i,j} = 1\}$ and the negative guidance from the remaining generations, $g_i^- \sim \{g_{i,j}|o_{i,j} = 0\}$.

## 3.3 ITERATIVE GUIDANCE OPTIMIZATION

As white-box LLMs like LLaMA are pre- and post-trained for general purposes, they may struggle to fulfill the specific tasks required by the controller. Additionally, there may be discrepancies between what the controller considers "good" guidance and what the generator interprets as "good" guidance. To this end, the guidance generated by the white-box LLM controller needs further optimization to enhance the performance of the black-box LLM generator.

**Supervised Fine-Tuning for Behavior Cloning.** To quickly initialize the controller's policy, we adopt the concept of behavior cloning (BC) from reinforcement learning, which involves learning an initial policy by imitating the actions of an expert agent. This is typically achieved through supervised learning on a set of curated instruction-completion pairs for LLMs. We leverage the capabilities of more advanced models, such as `GPT-4` (OpenAI, 2024a), to generate the desired guidance for the black-box LLMs on a small set of samples. This data is then used to perform supervised fine-tuning (SFT) on the white-box LLM controller as an initial warm-up step:

$$\mathcal{L}_{\text{SFT}} = -\mathbb{E}_{(x,g)\sim\mathcal{D}_{\text{SFT}}}\left[\sum_{l=1}^{L} \log f_\theta(g_l|g_{<l}, x)\right]. \tag{4}$$

Through this SFT process, the white-box LLM controller begins to acquire the capability to effectively guide the subsequent black-box LLM. It can then be utilized to generate high-quality guidance for further optimization steps.

**Direct Guidance Optimization.** By allowing the warmed-up white-box LLM controller to interact with the black-box LLM environment over multiple turns, we can curate a dataset containing

both "good" and "bad" guidance pairs from `Matryoshka`'s intermediate generations. Following reinforcement learning with human feedback (RLHF) (Bai et al., 2022; Ouyang et al., 2022; Ziegler et al., 2019), we obtain relative feedback by comparing pairs of generated guidance that share the same prompt. The preference signal is modeled using the Bradley-Terry model (Bradley & Terry, 1952). Given an input $x$ and a generated guidance pair $(g^+, g^-)$, the model specifies the probability of $g^+$ being chosen over $g^-$ as:

$$p(g^+ \succ g^- | x) = \frac{\exp\left(u\left(g^+; x\right)\right)}{\exp\left(u\left(g^+; x\right)\right) + \exp\left(u\left(g^-; x\right)\right)} = \sigma\left(u\left(g^+; x\right) - u\left(g^-; x\right)\right), \quad (5)$$

where $\sigma(x) = \frac{e^x}{(e^x+1)}$ is the logistic function. This formulation allows us to access sequence-level preferences to optimize the intermediate guidance generated by the white-box LLM controller. Following Rafailov et al. (2024b), we establish a connection between the white-box LLM controller and its associated optimal policy. Specifically, we consider the following KL-regularized planning problem with respect to a reference policy $\pi_{\text{ref}}$:

$$\max_\theta \mathbb{E}_{x \sim \mathcal{X}} \mathbb{E}_{g \sim \pi_\theta(\cdot|x)} \left[ u\left(x, g\right) - \eta^{-1} \mathbb{D}_{\text{KL}} \left[ \pi_\theta\left(g|x\right) || \pi_{\text{ref}}\left(g|x\right) \right] \right]. \quad (6)$$

The optimization problem above has a closed-form solution. For any guidance $g$, the optimal policy $\pi^*$ is given by:

$$\pi^*(g|x) \propto \pi_{\text{ref}}(g|x) \exp(\eta u(x, g)). \quad (7)$$

This leads us to the direct preference optimization (DPO) loss for optimizing the intermediate guidance generated by the white-box LLM controller:

$$\mathcal{L}_{\text{DPO}} := \mathbb{E}_{(x,g^+,g^-) \sim \mathcal{D}} \left[ -\log \sigma \left( \eta^{-1} \left[ \log \left( \frac{\pi_\theta(g^+|x)}{\pi_{\text{ref}}(g^+|x)} \right) - \log \left( \frac{\pi_\theta(g^-|x)}{\pi_{\text{ref}}(g^-|x)} \right) \right] \right) \right]. \quad (8)$$

**Iterative Guidance Optimization.** In the previous sections, we discussed sampling positive and negative intermediate generations through multi-turn interactions with the environment. However, an imbalance between positive and negative samples may arise, leading to overfitting on simplistic patterns and hindering the self-improvement of the white-box LLM controller. To address this issue, we propose an iterative guidance optimization method (Figure 4) that interleaves data sampling and training steps. We begin by initializing the model with parameters $\theta^{(0)} = \theta$ without any prior training and sample an initial dataset $\mathcal{D}^{(0)}$ as introduced in Section 3.2. At the $m$-th iteration, we have the optimized model $\theta^{(m)}$. Following STaR (Zelikman et al., 2022), we enhance the model's generation for the next iteration by bootstrapping the dataset. This involves combining the previous datasets with new samples generated by the current model $\{g_{i,0}^{(m)}, g_{i,1}^{(m)}, \cdots, g_{i,K \times T}^{(m)}\} \sim f_{\theta^{(m)}}(x)$:

$$\begin{cases} \mathcal{D}_+^{(m)} = \{g_{i,j}^{(m)} | o_{i,j}^{(m)} = 1\} \cup \mathcal{D}_+^{(m-1)}, \\ \mathcal{D}_-^{(m)} = \{g_{i,j}^{(m)} | o_{i,j}^{(m)} = 0\} \cup \mathcal{D}_-^{(m-1)}, \end{cases} \quad (9)$$

In the $t$-th iteration, we construct the training dataset $\mathcal{D}^{(t)}$ for DPO by sampling positive and negative pairs for each question. When training the model for the next iteration $\theta^{(m+1)}$, we update the reference policy to be the model from the previous iteration, $\pi_\theta^{(m)}(g|x)$. Consequently, the training objective for iterative guidance optimization of the white-box LLM controller becomes:

$$\mathcal{L}_{\text{IGO}} := \mathbb{E}_{(x,g^+,g^-) \sim \mathcal{D}} \left[ -\log \sigma \left( \eta^{-1} \left[ \log \left( \frac{\pi_\theta^{(m+1)}(g^+|x)}{\pi_\theta^{(m)}(g^+|x)} \right) - \log \left( \frac{\pi_\theta^{(m+1)}(g^-|x)}{\pi_\theta^{(m)}(g^-|x)} \right) \right] \right) \right].$$

## 4 EXPERIMENTS

In this section, we present comprehensive experiments on a diverse set of complex, long-horizon tasks to demonstrate the enhanced capabilities of black-box LLMs using `Matryoshka`.

### 4.1 EXPERIMENTAL SETUP

**Tasks and Datasets.** We consider three types of tasks in experiments, each targeting a distinct capability of black-box LLMs: (1) *LaMP* (Salemi et al., 2023) for personalization capabilities, (2) *GSM8K* (Cobbe et al., 2021) for reasoning capabilities, and (3) *ALFWorld* (Shridhar et al., 2020) for planning capabilities. Dataset details are available in Appendix C.

**Baselines.** We consider the following baselines: (1) *Baselines in personalization*, we consider both one-stage and two-stage personalization models, including Profile-Augmented Generation (PAG) (Richardson et al., 2023) and Retrieval-Augmented Generation (RAG) (Salemi et al., 2023). (2) *Baselines in reasoning*, we include Chain-of-Thoughts (CoT) (Wei et al., 2022), Least-to-Most (Zhou et al., 2023), Program-Aided Language Models (PAL) (Gao et al., 2023), and PAL$_{\text{Self-Debug}}$ (Chen et al., 2023). (3) *Baselines in planning*, we mainly compare Matryoshka with BUTLER (Shridhar et al., 2020), ReAct (Yao et al., 2023), Reflextion (Shinn et al., 2023), and AdaPlanner (Sun et al., 2024a). Baseline details can be found in Appendix D.

**Evaluation Metrics.** For the personalization tasks, consistent with the evaluation metrics specified in LaMP (Salemi et al., 2023), we use accuracy (*Acc*) and F1 score (*F1*) for the classification tasks in LaMP-2N and LaMP-2M. For the ordinal multi-class classification task in LaMP-3, we employ mean absolute error (*MAE*) and root mean squared error (*RMSE*). To comprehensively evaluate the personalized text generation tasks in LaMP-4 and LaMP-5, we report ROUGE-1 (*R-1*), ROUGE-L (*R-L*), and *BLEU* scores. For the math reasoning task, we assess the models based on the *accuracy* of obtaining the final correct answer. For the planning task, consistent with previous works (Sun et al., 2024a), we evaluate performance using the *success rate* (%). The success rate is calculated as the number of successful episodes divided by the total number of episodes. In ALFWorld, an episode is considered a failure if the task remains unsolved after executing 50 actions, which is the maximum allowed number of actions per episode.

**Implementations.** For the white-box LLM controller, we utilize `LLaMA-3-8B-Instruct` as the backbone language model. In the black-box LLM environment, our experiments employ `gpt-4o-mini` for personalization tasks in LaMP, and `gpt-3.5-turbo` for reasoning and planning tasks in GSM8K and ALFWorld, respectively. All experiments with GPTs are conducted using the Microsoft Azure OpenAI service. Please refer to Appendix E for implementation details.

### 4.2 PERSONALIZATION: LaMP

**Main Results.** Table 1 summarizes the primary experimental results on the LaMP dataset. Our proposed method, Matryoshka, consistently outperforms or matches other state-of-the-art baselines, highlighting its efficacy of advancing black-box LLMs in personalization. For classification tasks, Matryoshka achieves an accuracy of 0.832 on LaMP-2N and 0.535 on LaMP-2M, surpassing other baselines by a significant margin. For generation tasks, Matryoshka also attains over a 25% improvement in BLEU score on LaMP-4. These results demonstrate the effectiveness of Matryoshka in both classification and generative personalization tasks. Furthermore, Matryoshka has the potential to be enhanced with RAG, combining intermediate generations with the retrieved user history data to improve performance further.

**Ablation Studies.** In our ablation studies, we compare our proposed method, Matryoshka, with a baseline lacking Intermediate Guidance Optimization (IGO) in Table 1. Using the same black-box model (`gpt-4o-mini`), our optimized white-box controller consistently and significantly outperformed the original `LLaMA-3-8B-Instruct`. These results demonstrate the effectiveness of IGO in enhancing the white-box controller to generate more informative and higher-quality intermediate outputs, thereby guiding the black-box model toward better final answers.

**Plug-and-Play.** Matryoshka can seamlessly apply the optimized white-box controller, `LLaMA-3-8B-Instruct`, to other black-box models in a plug-and-play manner without additional training costs. We further utilize this well-tuned white-box controller as a plug-in to integrate with black-box models such as `gpt-3.5-turbo` and `gemini-1.5-flash`. Table 2 presents the plug-and-play results. The experimental results show that our well-tuned controller consistently outperforms other baselines. Specifically, on LaMP-3 and LaMP-4, our plug-in surpasses other baselines by a large margin, demonstrating effectiveness across both classification and generation

| Dataset ($\rightarrow$) | LaMP-1 | | LaMP-2N | | LaMP-2M | | LaMP-3 | | LaMP-4 | | |
|---|---|---|---|---|---|---|---|---|---|---|---|
| Method ($\downarrow$) | Acc. $\uparrow$ | F-1 $\uparrow$ | Acc. $\uparrow$ | F-1 $\uparrow$ | Acc. $\uparrow$ | F-1 $\uparrow$ | MAE $\downarrow$ | RMSE $\downarrow$ | R-1 $\uparrow$ | R-L $\uparrow$ | BLEU $\uparrow$ |
| `gpt-4o-mini` (OpenAI, 2024b) | 0.514 | 0.513 | 0.655 | 0.473 | 0.413 | 0.325 | 0.371 | 0.673 | 0.132 | 0.116 | 0.992 |
| RAG (k=1) (Salemi et al., 2023) | 0.626 | 0.624 | 0.733 | 0.539 | 0.444 | 0.378 | 0.311 | 0.631 | 0.141 | 0.126 | 1.296 |
| RAG (k=4) (Salemi et al., 2023) | 0.632 | 0.632 | 0.792 | 0.611 | 0.502 | 0.430 | **0.272** | **0.579** | 0.161 | 0.146 | 2.953 |
| PAG (Richardson et al., 2023) | 0.624 | 0.624 | 0.775 | 0.559 | 0.496 | 0.443 | 0.316 | 0.645 | 0.143 | 0.130 | 1.968 |
| Matryoshka | **0.640** | **0.640** | **0.832** | **0.614** | **0.535** | **0.475** | 0.282 | 0.588 | **0.171** | **0.157** | **4.144** |
| w/o IGO | 0.611 | 0.611 | 0.807 | 0.575 | 0.496 | 0.432 | 0.311 | 0.636 | 0.131 | 0.120 | 1.341 |

Table 1: Main experimental results on the personalization task using the LaMP benchmark. For all baselines, including `Matryoshka`, we utilize `gpt-4o-mini` as the black-box LLM generator. R-1 and R-L refer to ROUGE-1 and ROUGE-L, respectively. $k$ denotes the number of items retrieved. An upward arrow ($\uparrow$) indicates that higher values are preferred, whereas a downward arrow ($\downarrow$) signifies that lower values are better. The best score and second-best score for each task are emphasized in **bold** and underlined, respectively. IGO represents Intermediate Guidance Optimization. Notations are consistent across tables.

| Dataset ($\rightarrow$) | LaMP-1 | | LaMP-2N | | LaMP-2M | | LaMP-3 | | LaMP-4 | | |
|---|---|---|---|---|---|---|---|---|---|---|---|
| Method ($\downarrow$) | Acc. $\uparrow$ | F-1 $\uparrow$ | Acc. $\uparrow$ | F-1 $\uparrow$ | Acc. $\uparrow$ | F-1 $\uparrow$ | MAE $\downarrow$ | RMSE $\downarrow$ | R-1 $\uparrow$ | R-L $\uparrow$ | BLEU $\uparrow$ |
| `Matryoshka (4o-mini)` | **0.640** | **0.640** | **0.832** | **0.614** | **0.535** | **0.475** | **0.282** | **0.588** | **0.171** | **0.157** | **4.144** |
| `gpt-3.5-turbo` | 0.590 | 0.589 | 0.790 | 0.594 | 0.399 | 0.325 | 0.357 | 0.693 | 0.166 | 0.150 | 3.433 |
| PAG (gpt-3.5) | 0.590 | 0.589 | 0.790 | 0.594 | 0.399 | 0.325 | 0.357 | 0.693 | 0.166 | 0.150 | 3.433 |
| Plug-and-play (gpt-3.5) | **0.594** | **0.593** | **0.798** | **0.609** | 0.469 | 0.412 | **0.286** | **0.599** | **0.176** | **0.161** | **4.222** |
| w/o IGO (gpt-3.5) | 0.585 | 0.585 | 0.790 | 0.608 | **0.472** | **0.425** | 0.334 | 0.670 | 0.160 | 0.147 | 3.015 |
| `gemini-1.5-flash` | 0.518 | 0.510 | 0.700 | 0.498 | 0.368 | 0.279 | 0.546 | 0.825 | 0.135 | 0.113 | 1.494 |
| Plug-and-play (gemini) | **0.573** | **0.565** | **0.825** | **0.615** | 0.504 | **0.418** | **0.298** | **0.614** | **0.183** | **0.170** | **5.002** |
| w/o IGO (gemini) | 0.568 | 0.561 | 0.811 | 0.602 | **0.505** | 0.411 | 0.365 | 0.715 | 0.164 | 0.150 | 3.439 |

Table 2: Plug-and-Play experimental results for `gpt-3.5-turbo` and `gemini-1.5-flash` across the LaMP benchmark. We employ `Matryoshka` pre-trained on `gpt-4o-mini` as the white-box LLM controller.

tasks. The effectiveness of `Matryoshka` in plug-and-play scenarios arises from the generalization capability of intermediate guidance, which can benefit different black-box LLMs.

**Effects of Profile Count.** To further investigate the effect of user profile count on the generation of intermediate outputs, we analyze performance across different numbers of profiles per user. Figure 5 presents the accuracy and ROUGE-L curves separately for LaMP-2M and LaMP-4, with the x-axis representing the total number of profiles per user (*e.g.*, "0-20" indicates users with 0 to 20 profiles). We compared the results of our proposed method, `Matryoshka`, and PAG, utilizing the white-box controller `Llama-3-8B-Instruct` and

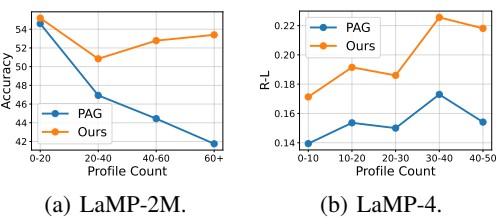

(a) LaMP-2M.   (b) LaMP-4.

Figure 5: Effect of number of history per user in LaMP-2M and LaMP-4.

the black-box model `gpt-4o-mini`. On LaMP-2M, as the profile count increases, PAG's performance significantly deteriorates, whereas `Matryoshka` maintains stable performance and surpasses PAG by an increasing margin. For LaMP-4, both `Matryoshka` and PAG exhibit similar trends, but `Matryoshka` consistently outperforms PAG by a substantial and steady margin. These results demonstrate the efficacy of IGO in enhancing the summarization capabilities of the black-box controller, especially when dealing with varying and large amounts of profiles.

## 4.3 REASONING: GSM8K

Table 3 presents the main results on the GSM8K dataset. We employ a three-shot prompt design across all baselines, including ours. $\text{PAL}_{\text{Self-Debug}}$ refers to the addition of close-loop refinement to PAL during the inference stage. Our method consistently outperforms all baselines across the dataset, surpassing the strongest baseline, $\text{PAL}_{\text{Self-Debug}}$, by a margin of 4.2% when using the base LLM.

This improvement stems from the optimized intermediate guidance generated by `Matryoshka`. Conditioned on this guidance, `Matryoshka` enables the black-box LLM to generate long-horizon solutions to solve the tasks. Similar to LaMP, `Matryoshka` trained with `gpt-3.5-turbo` can be seamlessly applied to other black-box models for solving mathematical problems on GSM8K without additional training costs. Notably, `Matryoshka` learns high-level planning abilities without focusing on specific details, which broadens its applicability.

| Dataset ($\rightarrow$) | GSM8K | | GSM-HARD | |
|---|---|---|---|---|
| **Methods** ($\downarrow$) | `gpt-3.5` | `4o-mini` | `gpt-3.5` | `4o-mini` |
| CoT | 0.809 | 0.932 | 0.406 | 0.500 |
| Least-to-Most | 0.811 | 0.908 | 0.425 | 0.498 |
| PAL | 0.802 | 0.920 | 0.638 | 0.748 |
| PAL$_{\text{Self-Debug}}$ | 0.864 | 0.943 | 0.701 | 0.774 |
| `Matryoshka` | **0.911** | **0.956** | **0.738** | 0.779 |
| w/o IGO | 0.896 | 0.954 | 0.729 | **0.780** |

Table 3: Accuracy on the mathematical reasoning task using the GSM8K dataset.

### 4.4 PLANNING: ALFWORLD

**Main Results.** `Matryoshka` consistently outperforms existing baselines, achieving state-of-the-art performance with an overall success rate of 95.52% on ALFWorld tasks (Table 4). This superior performance indicates that `Matryoshka` effectively generates plans to guide the task execution of the black-box model, enhancing its ability to interact with the environment. Furthermore, we observe that `Matryoshka` exhibits superior performance compared to both the untuned white-box model (w/o Guidance Optimization) and the white-box models trained with fewer rounds of Intermediate Guidance Optimization (w/o 1$^{\text{st}}$/2$^{\text{nd}}$-round IGO). As the number of IGO training rounds increases, `Matryoshka`'s performance on ALFWorld correspondingly improves, ultimately raising the success rate from 81.34% to 95.52%. These results underscore the efficacy of the IGO training employed in `Matryoshka`.

**Ablation Studies on Sample Efficiency.** Figure 6(a) illustrates the relationship between success rate (%) and the proportion of training data used to optimize the controller. In the ALFWorld environment, `Matryoshka` achieves an accuracy of 94.78% using only one-quarter of the training data, surpassing the best-performing baseline, AdaPlanner, by 6.7%. This demonstrates the sample efficiency of `Matryoshka` in achieving high performance with limited training data. This study demonstrates that `Matryoshka` significantly reduces

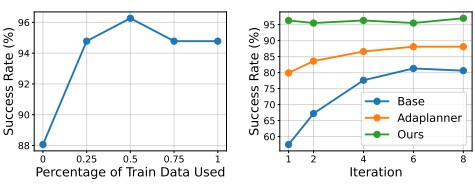

(a) Sample Efficiency. (b) Number of Turns $M$.

Figure 6: Success rate (%) *w.r.t* number of (a) training samples and (b) interaction turns.

the reliance on high-quality task planning samples and expert trajectories, thereby providing a more resource-efficient solution.

**Effect of Number of Interaction Turns $M$.** We evaluate the performance of `Matryoshka` against the vanilla `LLaMA3-8B-Instruct` and AdaPlanner on ALFWorld, varying the number of closed-loop iterations $M$ during the inference phase. As illustrated in Figure 6(b), following DPO training, `Matryoshka` achieves an accuracy exceeding 95% in the open-loop inference setting ($M$=1), significantly surpassing both AdaPlanner and `LLaMA3-8B-Instruct`. Furthermore, during an 8-iteration closed-loop inference, `Matryoshka` maintains the highest accuracy of 97%. These findings indicate that `Matryoshka` is capable of generating exceptionally high-quality plans, enabling the GPT model serving as the task executor to interact with the environment and complete tasks successfully without requiring closed-loop refinement.

## 5 RELATED WORKS

**Black-Box LLMs Generation Enhancement.** Existing approaches aiming to enhance the generation capabilities of black-box LLMs can be broadly categorized into two groups: (1) *ICL-* and (2) *adapter-based* methods. ICL-based methods (Sun et al., 2024a; Tan et al., 2024a; Zhuang et al., 2024) are designed to augment the original query with carefully crafted instructions or meticulously constructed few-shot demonstrations to guide the model. While this enables the black-box LLM to exhibit specific

| Methods (↓)  Tasks (→) | Pick | Clean | Heat | Cool | Examine | Pick Two | All (134 tasks) |
|---|---|---|---|---|---|---|---|
| BUTLER (Shridhar et al., 2020) | 46.00 | 39.00 | 74.00 | **100.00** | 22.00 | 24.00 | 37.00 |
| ReAct (Yao et al., 2023) | 37.50 | 64.52 | 69.57 | 42.86 | 38.89 | 17.65 | 47.76 |
| Reflexion (Shinn et al., 2023) | 50.00 | 41.94 | 65.22 | 52.38 | 66.67 | 47.06 | 52.99 |
| AdaPlanner (Sun et al., 2024a) | **100.00** | **93.55** | 78.26 | 95.24 | 66.67 | **88.24** | 88.06 |
| Matryoshka | **100.00** | **93.55** | **100.00** | 90.48 | **100.00** | **88.24** | **95.52** |
| w/o 2$^{nd}$-round IGO | 100.00 | 93.55 | 100.00 | 100.00 | 83.33 | 88.24 | 94.78 |
| w/o 1$^{st}$, 2$^{nd}$-round IGO | 100.00 | 93.55 | 86.96 | 95.24 | 55.56 | 88.24 | 88.06 |
| w/o Guidance Optimization | 100.00 | 93.55 | 91.30 | 85.71 | 11.11 | 88.24 | 81.34 |

Table 4: Success rate (%) across six planning tasks from AlfWorld. For all baselines, including `Matryoshka`, we utilize `gpt-3.5-turbo` as the black-box LLM generator.

capabilities or adhere to particular directives, these methods require significant human effort in prompt engineering and result in prompts that are rigid and static. Adapter-based methods (Sun et al., 2024b; Shi et al., 2024; Zhuang et al., 2024) follow a best-of-N selection evaluation paradigm (Lightman et al., 2023). Given a problem, adapter-based methods generate $N$ candidate solutions from the generator and subsequently evaluate them using a lightweight adapter to identify the highest-scoring solution as the final answer. However, such methods are heavily dependent on the generative capabilities of the black-box LLM, which may result in selecting a suboptimal candidate as *the best of a bad bunch*.

**Small LMs Drive LLMs Generation.** SuperICL (Xu et al., 2023) incorporates outputs from smaller language models (LMs) as complementary information for input queries, integrating them into the context provided to black-box LLMs. However, these smaller LMs are fixed and can only support classification tasks that rely on label predictions with associated confidence scores. HYDRA (Zhuang et al., 2024) is a retrieval-augmented generation framework that trains a BERT-sized reranker to reorder retrieved passages to better cater to user-specific requirements. Nevertheless, these methods apply only discrete optimization on the prompt through reranking and selection of few-shot demonstrations, which limits the potential improvements achievable via prompt engineering.

**Reinforcement Learning for Prompt Optimization.** As LLMs scale, new capabilities emerge, enabling models to learn tasks efficiently through a few in-context demonstrations. To harness these capabilities, several approaches have been proposed to leverage reinforcement learning for improved prompt generation, enhancing LLM performance. RLPrompt (Deng et al., 2022) introduces an RL-based framework for generating optimal prompts via black-box optimization. Similarly, TEMPERA (Zhang et al., 2023) formulates prompt optimization as test-time prompt editing, using RL to efficiently explore the editing space. BDPL (Diao et al., 2023) further advances this by proposing a variance-reduced policy gradient algorithm to estimate gradients of parameters in the categorical distribution of each discrete prompt. However, these methods primarily focus on classification tasks, where gradient estimation is straightforward, limiting their applicability to more complex generation tasks requiring long-horizon solutions.

## 6  Conclusion and Future Work

We introduced `Matryoshka`, a lightweight white-box LLM controller designed to augment the capabilities of large-scale black-box LLMs across a wide range of complex tasks, including reasoning, planning, and personalization. By leveraging a controller-generator framework with environmental feedback, `Matryoshka` effectively decomposes complex tasks and guides black-box LLMs through intermediate guidance. Through policy gradient optimization, `Matryoshka` exhibits a self-improving nature that continually enhances LLM capabilities via multi-turn guidance optimization. Extensive experiments on three diverse datasets demonstrate its effectiveness in steering black-box LLMs for long-horizon tasks without requiring access to model parameters or output probabilities. Compared to the best-performing state-of-the-art baselines, `Matryoshka` achieves average improvements of 3.19% in reasoning tasks, 7.46% in planning tasks, and 5.82% in personalization tasks. These results underscore the potential `Matryoshka` as a transparent and scalable solution, enabling white-box LLMs to drive black-box LLMs in complex problem-solving. Future work could extend `Matryoshka` to tackle more complex applications requiring long-horizon generation and reasoning, such as solving software engineering problems and proving mathematical theorems. Additionally, the controller component of `Matryoshka` could be developed into a self-enhancing mechanism or a universal controller applicable to a wide range of real-world applications.

## 7    REPRODUCIBILITY STATEMENT

The datasets utilized in this study are all publicly available, including LaMP for the personalization task, GSM8K and GSM-Hard for the reasoning task, and ALFWorld for the planning task. Detailed descriptions of these datasets and their corresponding tasks are provided in Appendix C. In Appendix D, we outline the baselines used for comparison and describe the experimental setup. Appendix E offers an in-depth explanation of the main experiments, including hardware and software configurations, hyperparameter settings, and step-by-step procedures for the three tasks. Additionally, Appendix F presents case studies for each task, demonstrating the superior performance of our method compared to the baselines in a more intuitive manner. Appendix G details the prompts used for each task. The implementation of `Matryoshka` is provided in the supplementary materials and will be released publicly available on GitHub upon acceptance.

## 8    ETHICS STATEMENT

We strictly followed the data usage guidelines for interactions with Microsoft Azure's OpenAI API and Gemini API service. Although our research relied solely on publicly available datasets, we took extra precautions to minimize any potential risk of information leakage. Specifically, we opted out of the human review process by completing and submitting the Azure OpenAI Additional Use Case Form[2]. This proactive measure highlights our commitment to maintaining the highest data privacy standards and ethical research practices, especially concerning personalization tasks.

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

## A   LIMITATIONS AND BROADER IMPACTS

### A.1   LIMITATIONS

In this study, we propose a modular framework, `Matryoshka`, that leverages a lightweight white-box LLM controller to enhance the capabilities of black-box LLMs. Despite its effectiveness, we have identified several potential limitations of `Matryoshka`:

**Malign Usage.** Since `Matryoshka` employs a white-box LLM controller to augment black-box LLMs, there are notable risks to consider. Malicious actors could exploit this approach to engineer harmful capabilities or generate toxic content for training purposes. While black-box LLMs are designed to resist producing such content, our controller could be misused to manipulate these models into generating undesirable outputs. Furthermore, there is a risk that the intermediate guidance produced by our controller could be exploited to extract sensitive information from black-box LLMs, potentially facilitating jailbreaking or other targeted attacks.

**Data Privacy.** `Matryoshka` preserves the confidentiality of training data by avoiding third-party API sharing, thereby safeguarding the integrity of training samples during the enhancement process of black-box LLMs. However, when applied to personalization tasks, it is important to recognize that retrieved historical records or the queries themselves may inadvertently contain sensitive information, potentially risking unintended disclosure of private data.

### A.2   BROADER IMPACTS

**Potential Positive Societal Impacts.** The proposed `Matryoshka` framework addresses a critical challenge in consistently enhancing the capabilities of black-box LLMs for long-horizon tasks with broad scopes. By improving reasoning, planning, and personalization, `Matryoshka` can deliver significant benefits across various domains. For instance, it can provide insights into complex theorems, advance industrial automation, and offer more personalized interactions for end users. Overall, `Matryoshka` has the potential to facilitate more useful, relevant, and satisfying interactions, thereby improving productivity, decision-making, and quality of life. Moreover, `Matryoshka` operates without requiring access to the model weights of black-box LLMs, making the technology accessible to a wide range of off-the-shelf LLM APIs and enabling seamless integration into diverse use cases. By leveraging existing LLMs, `Matryoshka` can be readily adopted by researchers, developers, and organizations, accelerating the development and deployment of advanced language models in real-world applications.

**Potential Negative Societal Impacts.** Enhancing black-box LLMs through a small-scale white-box LLM introduces potential risks. One significant concern is the possibility of using the white-box model to jailbreak black-box LLMs, injecting malicious instructions or producing harmful content. This could lead to the spread of misinformation, hate speech, or other offensive materials, with severe consequences for individuals and society. Additionally, this approach poses a threat to user data privacy. Training the white-box model requires collecting and storing interaction data between the

black-box LLM and the environment, which could be improperly handled or misused, potentially compromising sensitive information.

## B   ADDITIONAL RELATED WORKS

**RLHF.** Proximal policy optimization (PPO) (Schulman et al., 2017) is the predominant deep reinforcement learning method used in RLHF, leading to significant successes in models like InstructGPT (Ouyang et al., 2022), ChatGPT (Achiam et al., 2023), and Gemini (Reid et al., 2024). However, applying PPO requires extensive effort and resources (Choshen et al., 2019; Engstrom et al., 2020; Tang et al., 2024), often beyond the scope of open-source capabilities. To simplify implementation and streamline the training process, recent works (Azar et al., 2024; Ethayarajh et al., 2024) have proposed direct preference learning algorithms following the DPO framework (Rafailov et al., 2024b). These algorithms bypass the reward modeling step and directly optimize carefully designed loss objectives on the preference dataset, hence the term direct preference learning.

**Self-Improvement Training.** Recent advances in self-improvement methods for language models fall broadly into two categories: (1) online fine-tuning approaches and (2) bootstrapping methods. Fine-tuning approaches aim to enhance models by adjusting their parameters based on additional data or objectives. Notable methods include Rejection Fine-Tuning (RFT) (Yuan et al., 2023), which augments the training set with correct completions; Alignment Fine-Tuning (AFT) (Wang et al., 2023), which introduces an alignment loss to increase the probabilities of correct chain-of-thoughts; Reinforced Fine-Tuning (ReFT) (Luong et al., 2024), which applies reinforcement learning to token prediction; and self-play (Chen et al., 2024), which iteratively refines the model using its own previous outputs. Bootstrapping methods, on the other hand, leverage the model's own generations to create new training data. Notable examples include Self-Taught Reasoner (STaR) (Wu et al., 2024), which iteratively samples high-quality data; Reinforcement and Self-Training (ReST) (Gulcehre et al., 2023) and its simplified version ReST$^{\text{EM}}$ (Singh et al., 2023), which alternate between data generation and reward-based optimization; and Verified Self-Taught Reasoner (V-STaR) (Hosseini et al., 2024), which combines self-training with outcome-based verification. Collectively, these approaches offer diverse strategies for enhancing model performance through targeted training and iterative refinement, highlighting the potential for self-improvement in language models.

## C   DATASET AND TASK DETAILS

### C.1   LAMP: PERSONALIZATION

We employ the Language Model Personalization (LaMP) benchmark (Salemi et al., 2023), an open-source benchmark specifically designed to train and evaluate the capability of language models in generating personalized content. LaMP encompasses a diverse set of tasks (with LaMP-2 comprising two tasks, LaMP-2N, and LaMP-2M), covering both personalized text classification and generation tasks. The dataset statistics are presented in Table 5 for a clear overview of its structure. Below are detailed descriptions of each task:

- **Task 1: Personalized Citation Identification (LaMP-1)**: A binary text classification task aimed at citation recommendation. The task assesses the language model's ability to identify a user's citation preferences. Given a user and their authored paper, the model predicts which of two candidate papers the user is more likely to cite. The user's profile contains titles and abstracts of their authored papers.

- **Task 2: Personalized News Categorization (LaMP-2N)**: A categorical text classification task that involves categorizing news articles into one of 15 categories based on a journalist's profile. Given an article written by a user, the model predicts its category using the user's history of articles and their categories.

- **Task 3: Personalized Movie Tagging (LaMP-2M)**: An ordinal text classification task focused on predicting one of 15 tags for a movie based on a user's tagging history. The task evaluates the model's ability to assign tags to a movie description using historical user-specific movie-tag pairs.

- **Task 4: Personalized Product Rating (LaMP-3)**: A text classification task that involves predicting product ratings, framed as a five-class problem. The model must predict a rating between one and five for a product review, using the user's past review and rating history. This task tests the model's ability to capture user-specific rating patterns.

- **Task 5: Personalized News Headline Generation (LaMP-4)**: A text generation task in which the model generates personalized news headlines for articles based on the author's past article-title pairs. The task assesses the model's ability to replicate the author's stylistic preferences when creating headlines.

LaMP-6 has been excluded because the dataset is not publicly available. Furthermore, Tasks 1, 2, and 3 above cover personalization classification tasks, Task 4 covers personalization rating tasks, and Task 5 covers personalization generation tasks. Therefore, the tasks we selected encompass all categories of tasks in the LaMP benchmark.

Table 5: Dataset statistics of five different personalization tasks (LaMP-1, 2N, 2M, 3, and 4) from the LaMP benchmark (Salemi et al., 2023).

| Task | Type | # Train | # Validation | # Test | Input Length | Output Length | # Profiles | # Classes |
|------|------|---------|--------------|--------|--------------|---------------|------------|-----------|
| LaMP-1 | Classification | 9682 | 2500 | 2500 | $51.40 \pm 5.72$ | - | $90.61 \pm 53.87$ | 2 |
| LaMP-2N | Classification | 5914 | 1052 | 1274 | $65.40 \pm 12.29$ | - | $306.42 \pm 286.65$ | 15 |
| LaMP-2M | Classification | 5073 | 1410 | 1557 | $92.39 \pm 21.95$ | - | $86.76 \pm 189.52$ | 15 |
| LaMP-3 | Classification | 20000 | 2500 | 2500 | $145.14 \pm 157.96$ | - | $188.10 \pm 129.42$ | 5 |
| LaMP-4 | Generation | 12527 | 1925 | 2376 | $30.53 \pm 12.67$ | $9.78 \pm 3.10$ | $287.16 \pm 360.62$ | - |

## C.2 REASONING: GSM8K

GSM8K (Cobbe et al., 2021) is a dataset focused on high school-level mathematical reasoning. The numerical reasoning tasks within this dataset typically consist of a descriptive scenario followed by a culminating question. Answering these questions requires performing multi-step mathematical calculations based on the context provided in the description.

## C.3 PLANNING: ALFWORLD

AlfWorld (Shridhar et al., 2020) is a comprehensive suite of synthetic, text-based environments set within a virtual household, featuring six distinct task types: *Pick*, *Clean*, *Heat*, *Cool*, *Examine*, and *Pick Two*. Each task presents a unique high-level objective (e.g., "put a vase in the safe") that requires the agent to navigate and interact with various objects or receptacles (e.g., *go to shelf 6*, *clean apple*). To accomplish the assigned task, the agent must execute a series of actions to achieve the specified goal. However, the challenge lies in the object's potential location - it could be in any of over 50 possible places within a given task instance - necessitating sequential exploration of each location by the agent. Consequently, the complete action sequence may encompass more than 50 discrete actions, posing a considerable challenge to the agent's capabilities and efficiency.

## D BASELINE DETAILS

### D.1 LAMP: PERSONALIZATION

We compare our proposed `Matryoshka` with several competitive baselines, encompassing both one-stage and two-stage methods. For all baseline approaches, we employ a consistent prompt template and utilize BM25 as the default retrieval mechanism across all experiments.

- `gpt-4o-mini` follows a zero-shot approach, directly answering the user query without leveraging the user's profile data.

- **RAG** combines the user's top retrieved history data with the input question as prompts for `gpt-4o-mini` to generate the final answer.

- **PAG** utilizes `gpt-4o-mini` to first generate a summary of the user's retrieved history data and then combines the summary with the input question as prompts for `gpt-4o-mini` to produce the final answer.

For our ablation study, we primarily compare `Matryoshka` with the following ablated baseline:

- **Matryoshka w/o IGO** utilizes the controller model `Llama-3-8B-Instruct` to first generate a summary of the user's retrieved history data. It then combines this summary with the input question as prompts for the environment model `gpt-4o-mini` to generate the final answer.

### D.2 GSM: REASONING

For all baselines, we employ `gpt-3.5-turbo` as the black-box model to facilitate the description of their processes with 3-shot prompt template. The ablated baselines primarily focus on problem decomposition, including **Matryoshka w/o IGO**. The remaining baselines for mathematical reasoning consist of **CoT** (Wei et al., 2022), **Least-to-Most** (Zhou et al., 2023), **PaL** (Gao et al., 2023), and **PAL**$_{\text{Self-Debug}}$ (Chen et al., 2023).

- **Matryoshka w/o IGO** first utilizes a vanilla `LLaMA3-8B-Instruct` to break down the problem into sub-questions, and then `gpt-3.5-turbo` provide solutions based on both the main problem and the decomposed sub-questions.
- **CoT** uses `gpt-3.5-turbo` to break the problem down into a series of intermediate reasoning steps that ultimately lead to the final answer.
- **PaL** utilizes `gpt-3.5-turbo` to interpret natural language problems and generate programs as intermediate reasoning steps, delegating the solution process to a runtime environment like a Python interpreter.
- **PAL**$_{\text{Self-Debug}}$ builds upon PaL by introducing a close-loop refinement during the inference phase. Specifically, if the code generated by PaL encounters issues during execution, `gpt-3.5-turbo` is instructed to reflect on the error and regenerate the code. The maximum number of reflections is set to 6.

### D.3 ALFWORLD: PLANNING

We compare `Matryoshka` with several strong baselines in the planning task, encompassing both one-stage and two-stage approaches. For all baselines, we employ `gpt-3.5-turbo` as the black-box model for task execution. The ablated baselines (two-stage) include **w/o Guidance Optimization**, **w/o** $1^{\text{st}}, 2^{\text{nd}}$**-round IGO**, and **w/o** $2^{\text{nd}}$**-round IGO**. Additional baselines (one-stage) include **BUTLER** (Shridhar et al., 2020), **ReAct** (Yao et al., 2023), **Reflexion** (Shinn et al., 2023), and **AdaPlanner** (Sun et al., 2024a).

- **Ablated baselines.** These approaches utilize a white-box model to provide a high-level plan for the task, while `gpt-3.5-turbo` generates the specific solution based on this plan. Specifically:
  - **w/o Guidance Optimization** refers to an untuned `LLaMA3-8B-Instruct`.
  - **w/o** $1^{\text{st}}, 2^{\text{nd}}$**-round IGO** indicates a `LLaMA3-8B-Instruct` model that has undergone supervised fine-tuning on a limited amount of training data.
  - **w/o** $2^{\text{nd}}$**-round IGO** denotes the `LLaMA3-8B-Instruct` model further trained using DPO on {positive, negative} pairs from the training set, building upon the supervised fine-tuned model.
- **BUTLER** (Shridhar et al., 2020) is an agent that initially learns to perform abstract tasks in TextWorld through Imitation Learning (IL) and subsequently transfers the acquired policies to embodied tasks in ALFWorld.
- **ReAct** (Yao et al., 2023) is a general paradigm that combines reasoning and acting with language models to solve diverse language reasoning and decision-making tasks.
- **Reflexion** (Shinn et al., 2023) employs verbal reinforcement to help agents learn from prior failures.

- **AdaPlanner** (Sun et al., 2024a) is a closed-loop planning method where the LLM plays two roles, planner and refiner. It leverages code-based prompting for precise planning and refinement.

## E IMPLEMENTATION DETAILS

### E.1 HARDWARE AND SOFTWARE

We conduct all black-box LLM enhancement experiments on CPU: AMD(R) EPYC(R) 7702 64-Core Processor@1.50GHz and GPU: NVIDIA A100-SXM4-80GB using Python 3.10.13.

### E.2 LaMP: PERSONALIZATION

#### E.2.1 ALGORITHM DETAILS

We formalize the personalization problem within the context of our proposed `Matryoshka` framework. Specifically, we employ the controller model `Llama-3-8B-Instruct` to analyze the user's retrieved history data and generate an informative and clear intermediate summary. This summary is then combined with the input question as prompts for the environment model `gpt-4o-mini` to derive the final answer. To enhance control capabilities, we utilize online DPO to optimize the controller model `Llama-3-8B-Instruct`.

During the interaction stage, we follow the aforementioned pipeline, leveraging the controller model `Llama-3-8B-Instruct` to generate various intermediate outputs. By interacting with the environment model `gpt-4o-mini`, we obtain intermediate generations paired with ground truth answers as corresponding observations. We then sample both positive and negative intermediate generations based on the quality of the final answer. For classification tasks such as LaMP-1 and LaMP-2, an intermediate generation is labeled as positive if the final answer exactly matches the ground truth, and vice versa. For generation tasks like LaMP-4, we rank the generations by their metric scores and select the top ones as positive and the bottom ones as negative.

To prevent overfitting and reward hacking, the interaction stage processes the entire training dataset once for all personalization tasks. We sample at most two contrastive pairs for each training data point. In cases where no positive generation exists for some data points, we utilize a more powerful model, such as `gpt-4o`, to produce several strong intermediate generations, thereby increasing the likelihood of obtaining positive samples. We employ LoRA (Low-Rank Adaptation), a parameter-efficient method, to update the controller model `Llama-3-8B-Instruct`. LoRA is well-suited for personalization tasks, allowing efficient and effective optimization of the controller model. We utilize DPO for optimization.

#### E.2.2 HYPERPARAMETER CONFIGURATIONS

We set the maximum sequence length for generated solutions to 512 tokens across all tasks and scenarios. The controller model is `Llama-3-8B-Instruct`, while the environment model is `gpt-4o-mini` for the primary tasks and `gpt-3.5-turbo` for specific ablation studies. For each user, we retrieve the minimum of $k$ and the total number of user profiles as historical profiles. The value of $k$ varies by dataset: all profiles for LaMP-1, 120 for LaMP-2N, 150 for LaMP-2M, 30 for LaMP-3, and 50 for LaMP-4. These retrieved profiles are utilized in generating intermediate solutions. Comprehensive prompt templates and additional details are provided in Appendix G.

To prevent overfitting and reward hacking, we iterate through the entire training dataset only once. For each data point, we perform ten interactions, generating ten distinct intermediate solutions with a temperature setting of 1.0. If no positive samples are identified, we employ the more advanced `gpt-4o` to generate an additional five intermediate solutions, applying the same criteria to evaluate their positivity. Consequently, each data point results in at least ten intermediate generations. To further mitigate overfitting and reward hacking, we sample a maximum of two contrastive pairs per data point. The total number of contrastive pairs sampled during the interaction stage is as follows: 5410 for LaMP 1, 2850 for LaMP-2M, 2548 for LaMP-2N, 4320 for LaMP-3, and 12518 for LaMP-4, respectively.

During optimization, we train for two epochs per task using the following hyperparameters: LoRA rank to 8, LoRA $\alpha$ to 16, LoRA dropout to 0.05, learning rate to 1e-5, float type to bf16, max length to 8192, and label smoothing to 0.1. We utilize all the contrastive pairs sampled from the interaction stage for optimization. For all the experiments, we set all the random seeds to 42 for reproducibility consideration.

### E.3 GSM8K: REASONING

Following the PAL framework (Gao et al., 2023), we employ code-style LLM prompts to facilitate the conversion of mathematical problems into executable code, thereby augmenting the model's problem-solving capabilities. Unlike PAL, which directly translates mathematical problems into code, `Matryoshka` first assists GPT in decomposing the problem into more manageable sub-problems. This decomposition allows GPT to more effectively convert these simpler sub-problems into code, enhancing both the correctness and stability of the generated code. Additionally, since `Matryoshka` is responsible solely for high-level planning without engaging in low-level execution, we can train on the GSM8K dataset and evaluate on the GSM-Hard dataset. Both datasets comprise similar problem types, with GSM-Hard featuring more intricate numerical calculations.

In our experimental setup, we begin by randomly sampling 216 code-based solutions to mathematical problems from the GSM8K training set using `gpt-3.5-turbo-0125`. We then extract the planning components from these code blocks to perform supervised fine-tuning (SFT) on the LLaMA model, thereby equipping LLaMA with foundational planning capabilities for solving mathematical problems. The SFT training configuration mirrors that used for ALFWorld. Subsequently, LLaMA functions as the planner, generating breakdowns and planning solutions for each of the 7,473 problems in the GSM8K training set. Concurrently, GPT serves as the executor, producing executable code based on each problem and the corresponding plan provided by LLaMA.

During inference, consistent with our experiments on ALFWorld, we implement closed-loop refinement to enhance model performance. `Matryoshka` initially decomposes the mathematical problem into simpler sub-problems. The black-box model then generates corresponding code blocks for each sub-problem. If the execution of the generated code does not produce the expected answer or if execution issues arise, the error information is relayed back to the black-box model for reflection and iterative improvement. We restrict the number of reflection attempts to six; any problem that remains unresolved after these attempts is considered beyond the reasoning capabilities of the black-box model.

### E.4 ALFWORLD: PLANNING

Following AdaPlanner (Sun et al., 2024a), we employ a closed-loop planning approach for inference on ALFWorld. The primary distinction lies in `Matryoshka`'s responsibility for generating the high-level plan, while a black-box model, such as `gpt-3.5-turbo-0125`, handles low-level execution after comprehending both the problem and the high-level plan. Similar to AdaPlanner, we utilize code-style LLM prompts to enhance the black-box model's planning and interaction capabilities with the environment.

Our initial objective is to enhance LLaMA's planning ability on ALFWorld. To achieve this, we enable GPT to perform closed-loop high-level planning and low-level execution on 400 samples from ALFWorld's training set. From these runs, we selected 277 examples that successfully reached the goal state and extracted the planning components to fine-tune LLaMA using supervised learning. For the SFT, we set the learning rate to $2 \times 10^{-5}$, with a batch size of 64, and trained a LoRA module with a rank of 8, an alpha of 16, and a dropout rate of 0.05 over 3 epochs. After LLaMA acquires a foundational level of planning ability, we designate it as the planner and assign GPT as the executor. The two models then perform closed-loop inference on the ALFWorld training set, comprising 8,810 samples. Each sample is executed eight times, with successful runs labeled as positive samples and unsuccessful ones as negative samples. This process yields 4,844 unique {positive, negative} pairs, which are utilized for the first epoch of DPO training on LLaMA.

Subsequently, we repeat the data collection process on the ALFWorld training set using the DPO-trained model, gathering 1,586 samples. This reduction in samples occurs because, as `Matryoshka` becomes more capable post-DPO training, it generates a higher proportion of positive outcomes,

resulting in fewer {positive, negative} pairs. By aggregating all collected samples, we obtain a total of 6,430 pairs, which are then used to conduct the second epoch of DPO training on `Matryoshka`. This further enhances its planning capabilities and aligns them more closely with GPT's execution proficiency. Through this iterative DPO training approach, we observe that the high-level plans generated by LLaMA more effectively guide GPT's execution, leading to a higher success rate in ALFWorld tasks.

Additionally, during the inference stage, we maintain a closed-loop approach to bolster the model's performance. Specifically, the black-box model first generates a corresponding trajectory based on the task and the prompt provided by `Matryoshka`. If an error occurs during execution or the task remains incomplete after a predetermined number of steps, the black-box model reflects on its generated trajectory and the encountered error, subsequently regenerating a new trajectory. The model is permitted up to six reflection attempts. If it still fails to complete the task after these attempts, the task is deemed beyond the model's capabilities. This methodology effectively enhances the model's ability to interact with the environment and increases the likelihood of successfully completing tasks.

# F  CASE STUDIES

## F.1  ALFWORLD

We compare `Matryoshka` with both the `LLaMA3-8B-Instruct` model used as the controller and AdaPlanner (Sun et al., 2024a) in terms of performance across various tasks and environments. In `Matryoshka`, the controller model is trained based on `LLaMA3-8B-Instruct`, while the black-box executor employed is `gpt-3.5-turbo-0125`. Our results demonstrate that `Matryoshka` enhances the black-box executor's ability to interact with the environment, enabling it to successfully accomplish target tasks. In contrast, both AdaPlanner and the `LLaMA3-8B-Instruct` model commonly encounter issues such as initiating actions that the agent cannot complete and misjudging the agent's state. `Matryoshka` effectively mitigates these problems, resulting in more reliable task execution.

```
┌────────────────────── Task Definition ──────────────────────┐
 You are in the middle of a room. Looking quickly around you, you
 see a bed 1, a desk 2, a desk 1, a drawer 6, a drawer 5, a drawer
 4, a drawer 3, a drawer 2, a drawer 1, a garbagecan 1, a
 laundryhamper 1, a safe 1, a shelf 6, a shelf 5, a shelf 4, a shelf
 3, a shelf 2, and a shelf 1.

 Your task is to: examine the book with the desklamp.
└──────────────────────────────────────────────────────────────┘
```

```
┌────────────────────────── Matryoshka ──────────────────────────┐
 # Decomposition:
 General plan: I need to get a list of receptacles to find the book
 and take the book with me, then I get another list of receptacles
 to find the desklamp and turn it on.
 [Step 1] get a list of receptacles where a book is likely to appear.
 [Step 2] go to each receptacle in the list until seeing a book.
 [Step 3] take the book from the receptacle.
 [Step 4] get a list of receptacles where a desklamp is likely to appear.
 [Step 5] go to each receptacle in the list until seeing a desklamp.
 [Step 6] turn on the desklamp.
 [Step 7] go to the receptacle where the book is and turn on the desklamp.
 [Step 8] examine the book with the desklamp.

 # GPT Solution:
 def solution(agent, start_from=1):
     # General plan: I need to get a list of receptacles to find the
     book and take the book with me, then I get another list of
     receptacles to
     find the desklamp and turn it on.
     if start_from <= 1:
     print("[Step 1] get a list of receptacles where a book is
     likely to appear.")
└──────────────────────────────────────────────────────────────┘
```

```
        # Ask the assistant to sort the receptacles based on the
        likelihood
        of finding a book.
        answer = ask(f'Given a list of receptacles, please sort them in
        descending order based on the likelihood
        of finding a book in each of them. The list of receptacles is:
        {agent.receptacles}. You should directly return a Python list.')
        recep_to_check = literal_eval(answer)
        # Check if the returned list is not empty.
        assert recep_to_check, f'Error in [Step 1]: recep_to_check
        should not be empty. {agent.report()}'

    if start_from <= 2:
        print("[Step 2] go to each receptacle in the list until seeing
        a book.")
        for receptacle in recep_to_check:
            observation = agent.goto(receptacle)
            # Check if the receptacle is closed. If so, open it.
            if 'closed' in observation:
                observation = agent.open_receptacle(receptacle)
            # Check if a book is in/on the receptacle.
            if 'book' in observation:
                break
        # Check if a book is found in one of the receptacles.
        assert 'book' in observation, f'Error in [Step 2]: There is no
        book in/on {recep_to_check}. {agent.report()}'

    if start_from <= 3:
        print("[Step 3] take the book from the receptacle.")
        # Get the identifier of the book.
        answer = ask(f'From the observation, get the identifier of an
        object. For example, On the cabinet 1, you see a cloth 2, and a
        book 1. The identifier of cloth is 2. Now, {observation} The
        identifier of the book? Only Output a single number without any
        other words.')
        found_book = f'book {answer}'
        observation = agent.take(found_book, receptacle)
        # Check if the agent is holding the book.
        assert agent.holding == found_book, f'Error in [Step 3]: I
        cannot take {found_book} from the {receptacle}.
        {agent.report()}'

    if start_from <= 4:
        print("[Step 4] get a list of receptacles where a desklamp is
        likely to appear.")
        # Ask the assistant to sort the receptacles based on the
        likelihood of finding a desklamp.
        answer = ask(f'Given a list of receptacles, please sort them in
        descending order based on the likelihood of finding a desklamp
        in each of them. The list of receptacles is:
        {agent.receptacles}. You should directly return a Python list.')
        recep_to_check = literal_eval(answer)
        # Check if the returned list is not empty.
        assert recep_to_check, f'Error in [Step 4]: recep_to_check
        should not be empty. {agent.report()}'

    if start_from <= 5:
        print("[Step 5] go to each receptacle in the list until seeing
        a desklamp.")
        for receptacle in recep_to_check:
            observation = agent.goto(receptacle)
            # Check if the receptacle is closed. If so, open it.
            if 'closed' in observation:
                observation = agent.open_receptacle(receptacle)
            # Check if a desklamp is in/on the receptacle.
```

```
            if 'desklamp' in observation:
                break
        # Check if a desklamp is found in one of the receptacles.
        assert 'desklamp' in observation, f'Error in [Step 5]: There is
        no desklamp in/on {recep_to_check}. {agent.report()}'

    if start_from <= 6:
        print("[Step 6] turn on the desklamp.")
        # Get the identifier of the desklamp.
        answer = ask(f'From the observation, get the identifier of an
        object. For example, On the cabinet 1, you see a cloth 2, and a
        desklamp 1. The identifier of cloth is 2. Now, {observation}
        The identifier of the desklamp? Only Output a single number
        without any other words.')
        found_desklamp = f'desklamp {answer}'
        # Turn on the desklamp.
        observation = agent.turn_on(found_desklamp)
        # Check if the desklamp is turned on.
        assert 'turn on' in observation, f'Error in [Step 6]: I cannot
        turn on {found_desklamp} in/on {receptacle}. {agent.report()}'

    if start_from <= 7:
        print("[Step 7] go to the receptacle where the book is and turn
        on the desklamp.")
        observation = agent.goto(receptacle)
        # Check if the receptacle is closed. If so, open it.
        if 'closed' in observation:
            observation = agent.open_receptacle(receptacle)
        # Turn on the desklamp.
        observation = agent.turn_on(found_desklamp)
        # Check if the desklamp is turned on.
        assert 'turn on' in observation, f'Error in [Step 7]: I cannot
        turn on {found_desklamp} in/on {receptacle}. {agent.report()}'

    if start_from <= 8:
        print("[Step 8] examine the book with the desklamp.")
        # Examine the book with the desklamp.
        observation = f'You examine the {found_book} with the
        {found_desklamp}.'
        # Check if the book is examined with the desklamp.
        assert 'examine' in observation, f'Error in [Step 8]: I cannot
        examine the {found_book} with the {found_desklamp}.
        {agent.report()}'

Success
```

--- Vanilla LLaMA3-8B-Instruct ---

```
# Decomposition:
General plan: I need to get a list of receptacles to find the book
and take the book with me, then I get another list of receptacles
to find the desklamp and examine the book with it.
[Step 1] Get a list of receptacles where a book is likely to appear.
[Step 2] Go to each receptacle in the list until seeing a book.
[Step 3] Take the book from the receptacle.
[Step 4] Get a list of receptacles where a desklamp is likely to
appear.
[Step 5] Go to each receptacle in the list until seeing a desklamp.
[Step 6] Examine the book with the desklamp.

# GPT Solution:
def solution(agent, start_from=1):
    # General plan: I need to get a list of receptacles to find the
    book and take the book with me, then I get another list of
    receptacles to find the desklamp and examine the book with it.
```

```
1242        if start_from <= 1:
1243            print("[Step 1] Get a list of receptacles where a book is
1244            likely to appear.")
1245            # Ask the assistant to sort the receptacles based on the
1246            likelihood of finding a book.
1247            answer = ask(f'Given a list of receptacles, please sort
1248            them in descending order based on the likelihood of finding
1249            a book in each of them. The list of receptacles is:
1250            {agent.receptacles}. You should directly return a Python
1251            list.')
1251            recep_to_check = literal_eval(answer)
1252            # Check if the returned list is not empty.
1252            assert recep_to_check, f'Error in [Step 1]: recep_to_check
1253            should not be empty. {agent.report()}'
1254
1255        if start_from <= 2:
1256            print("[Step 2] Go to each receptacle in the list until
1257            seeing a book.")
1257            for receptacle in recep_to_check:
1258                observation = agent.goto(receptacle)
1259                # Check if the receptacle is closed. If so, open it.
1260                if 'closed' in observation:
1261                    observation = agent.open_receptacle(receptacle)
1262                # Check if a book is in/on the receptacle.
1263                if 'book' in observation:
1263                    break
1264            # Check if a book is found in one of the receptacles.
1265            assert 'book' in observation, f'Error in [Step 2]: There is
1266            no book in/on {recep_to_check}. {agent.report()}'
1267
1268        if start_from <= 3:
1268            print("[Step 3] Take the book from the receptacle.")
1269            # Get the identifier of the book from the observation.
1270            answer = ask(f'From the observation, get the identifier of
1271            an object. For example, On the cabinet 1, you see a cloth
1272            2, and a book 1. The identifier of cloth is 2. Now,
1273            {observation} The identifier of the book? Only Output a
1273            single number without any other words.')
1274            found_book = f'book {answer}'
1275            observation = agent.take(found_book, receptacle)
1276            # Check if the agent is holding the book.
1277            assert agent.holding == found_book, f'Error in [Step 3]: I
1277            cannot take {found_book} from the {receptacle}.
1278            {agent.report()}'
1279
1280        if start_from <= 4:
1281            print("[Step 4] Get a list of receptacles where a desklamp
1281            is likely to appear.")
1282            # Ask the assistant to sort the receptacles based on the
1283            likelihood of finding a desklamp.
1284            answer = ask(f'Given a list of receptacles, please sort
1285            them in descending order based on the likelihood of finding
1286            a desklamp in each of them. The list of receptacles is:
1287            {agent.receptacles}. You should directly return a Python
1287            list.')
1288            recep_to_check = literal_eval(answer)
1289            # Check if the returned list is not empty.
1290            assert recep_to_check, f'Error in [Step 4]: recep_to_check
1291            should not be empty. {agent.report()}'
1292
1293        if start_from <= 5:
1293            print("[Step 5] Go to each receptacle in the list until
1294            seeing a desklamp.")
1295            for receptacle in recep_to_check:
1295                observation = agent.goto(receptacle)
```

```
                    # Check if the receptacle is closed. If so, open it.
                    if 'closed' in observation:
                        observation = agent.open_receptacle(receptacle)
                    # Check if a desklamp is in/on the receptacle.
                    if 'desklamp' in observation:
                        break
            # Check if a desklamp is found in one of the receptacles.
            assert 'desklamp' in observation, f'Error in [Step 5]:
            There is no desklamp in/on {recep_to_check}.
            {agent.report()}'

        if start_from <= 6:
            print("[Step 6] Examine the book with the desklamp.")
            # Examine the book with the desklamp.
            observation = agent.examine(found_book,
            found_desklamp
            # Check if the book is being examined with the desklamp.
            assert 'examine' in observation, f'Error in [Step 6]: I
            cannot examine the book with the desklamp. {agent.report()}'

AttributeError: 'Agent' object has no attribute 'examine'}, in
step6: observation = agent.examine(found_book, found_desklamp)
```

```
┌─────────────── Adaplanner ───────────────┐

def solution(agent, start_from=1):
    # General plan: I need to find the book and examine it with the
    desklamp.

    if start_from <= 1:
        print("[Step 1] get a list of receptacles where a book is
        likely to appear.")
        # Ask the assistant to sort the receptacles based on the
        likelihood of finding a book.
        answer = ask(f'Given a list of receptacles, please sort
        them in descending order based on the likelihood of finding
        a book in each of them. The list of receptacles is:
        {agent.receptacles}. You should directly return a Python
        list.')
        recep_to_check = literal_eval(answer)
        # Check if the returned list is not empty.
        assert recep_to_check, f'Error in [Step 1]: recep_to_check
        should not be empty. {agent.report()}'

    if start_from <= 2:
        print("[Step 2] go to each receptacle in the list until
        finding a book")
        for receptacle in recep_to_check:
            observation = agent.goto(receptacle)
            # Check if the receptacle is closed. If so, open it.
            if 'closed' in observation:
                observation = agent.open_receptacle(receptacle)
            # Check if a book is in/on the receptacle.
            if 'book' in observation:
                break
        # Check if a book is found in the receptacle.
        assert 'book' in observation, f'Error in [Step 2]: There is
        no book in/on {recep_to_check}. {agent.report()}'

    if start_from <= 3:
        print("[Step 3] examine the book with the desklamp")
        # Find the desklamp to use for examining the book.
        answer = ask(f'Given the observation, find the identifier
        of the desklamp. For example, On the desk 1, you see a lamp
        1 and a book 2. The identifier of the lamp is 1. Now,
        {observation} What is the identifier of the desklamp? Only
```

```
            output a single number without any other words.')
            found_desklamp = f'desklamp {answer}'
            # Turn on the desklamp to examine the book.
            observation = agent.turn_on(found_desklamp)
            # Check if the desklamp is turned on.
            assert 'turn on' in observation, f'Error in [Step
            3]: Unable to turn on {found_desklamp} in/on {receptacle}.
            {agent.report()}'

AssertionError: Error in [Step 3]: Unable to turn on desklamp 0
in/on bed 1, in step3: assert 'turn on' in observation, f'Error in
[Step 3]: Unable to turn on {found_desklamp} in/on {receptacle}.
{agent.report()}'
```

## F.2 GSM-HARD

On the mathematical reasoning task, we compare `Matryoshka` with several strong baselines, including `LLaMA3-8B-Instruct` as the controller model, Chain-of-Thought (CoT) (Wei et al., 2022), and PaL (Gao et al., 2023). In `Matryoshka`, the controller model is trained based on `LLaMA3-8B-Instruct`, and the black-box model employed is `gpt-3.5-turbo-0125`. Our results indicate that common issues with the baselines include incorrect code execution order and difficulties in translating questions into code. In contrast, `Matryoshka` significantly enhances the black-box model's ability to solve mathematical problems through code generation.

—— **Task Definition** ——

```
There are 5915126 students in MarissaâĂŹs class. Each student started the
year with 10 pencils. After two months, 1/5 of the total pencils in
class were used. At the end of the year, only 1/3 of the remaining
pencils were left. How many pencils were left?

True Answer: 15773669.333333334
```

—— **Matryoshka** ——

```python
# Decomposition:
Let's break down this problem:
1. How many pencils were used in total?
2. How many pencils are left after using the total amount and then
losing 2/3 of the remaining pencils?

# GPT Solution:
def solution():
    """There are 5915126 students in MarissaâĂŹs class. Each student
    started the year with 10 pencils. After two months, 1/5 of the
    total pencils in class were used. At the end of the year, only 1/3
    of the remaining pencils were left. How many pencils were left?"""
    # Initialization of variables
    num_students = 5915126
    pencils_per_student = 10
    pencils_used_fraction = 1/5
    pencils_left_fraction = 1/3

    # How many pencils were used in total?
    total_pencils_initial = num_students * pencils_per_student
    pencils_used = total_pencils_initial * pencils_used_fraction

    # How many pencils are left after using the total amount and then
    losing 2/3 of the remaining pencils?
    pencils_remaining = total_pencils_initial - pencils_used
    pencils_left = pencils_remaining * (1 - pencils_left_fraction)
    result = pencils_left
    return result
```

```
Answer: 31547338.66666667, Wrong Answer

# Second-Turn Decomposition:
Let's break down this problem:
1. How many pencils were used in total?
2. How many pencils are left after using the total amount?
Note: We don't need to consider losing 2/3 of the remaining pencils in
the decomposition, as it's a part of the calculation for the total
pencils used.

# Second-Turn GPT Solution:
def solution():
    """There are 5915126 students in MarissaâĂŹs class. Each student
    started the year with 10 pencils. After two months, 1/5 of the
    total pencils in class were used. At the end of the year, only 1/3
    of the remaining pencils were left. How many pencils were left?"""
    # Initialization of variables
    num_students = 5915126
    pencils_per_student = 10
    pencils_used_fraction = 1/5
    pencils_left_fraction = 1/3

    # How many pencils were used in total?
    total_pencils_initial = num_students * pencils_per_student
    pencils_used = total_pencils_initial * pencils_used_fraction

    # How many pencils are left after using the total amount and then
    losing 2/3 of the remaining pencils?
    pencils_remaining = total_pencils_initial - pencils_used
    pencils_left = pencils_remaining * (1 - pencils_left_fraction)
    result = pencils_left
    return result

Answer: 15773669.333333332, Correct Answer
```

```
──────────────── Vanilla LLaMA3-8B-Instruct ────────────────
# Decomposition:
Let's break down this problem:
1. How many pencils were used in total?
2. How many pencils were left after the first two months?
3. How many pencils were left at the end of the year?

# GPT Solution:
def solution():
    """There are 5915126 students in MarissaâĂŹs class. Each student
    started the year with 10 pencils. After two months, 1/5 of the
    total pencils in class were used. At the end of the year, only 1/3
    of the remaining pencils were left. How many pencils were left?"""
    # Initialization of variables
    num_students = 5915126
    pencils_per_student = 10
    pencils_used_fraction = 2/5
    pencils_left_fraction = 1/3

    # How many pencils were used in total?
    total_pencils_initial = num_students * pencils_per_student
    pencils_used_total = total_pencils_initial * pencils_used_fraction

    # How many pencils were left after the first two months?
    pencils_remaining_after_two_months = total_pencils_initial -
    pencils_used_total

    # How many pencils were left at the end of the year?
    pencils_left = pencils_remaining_after_two_months *
```

```
        pencils_left_fraction
        result = pencils_left
        return result

Answer: 11830252.0, Wrong Answer
```

```
─────────────────────── PaL ───────────────────────
def solution():
    """There are 5915126 students in MarissaâĂŹs class. Each student started the year with 10
    num_students = 5915126
    pencils_per_student = 10
    total_pencils_initial = num_students * pencils_per_student

    pencils_used = total_pencils_initial // 5
    pencils_remaining = total_pencils_initial - pencils_used

    pencils_left_end_of_year = pencils_remaining // 3
    result = pencils_left_end_of_year
    return result

Answer: 15773669.0, Wrong Answer
```

```
─────────────────────── CoT ───────────────────────
First, we need to find out how many pencils were used after two months.
Since 1/5 of the total pencils were used, the number of pencils used is
1/5 * (5915126 * 10) = 1183025.2 pencils.

Next, we need to find out how many pencils were left at the end of the
year. Since only 1/3 of the remaining pencils were left, the number of
pencils left is 2/3 * (5915126 * 10 - 1183025.2) = 3156750.8 pencils.

Since we can't have a fraction of a pencil, we need to round down to
the nearest whole number. Therefore, the number of pencils left is
3156750 pencils. So the answer is 3156750.

Answer: 3156750, Wrong Answer
```

## F.3 LAMP

We utilize case study tables to illustrate the performance improvements achieved through the intermediate generations of the controller model. Specifically, we present detailed case studies for LaMP-2M, LaMP-2N, and LaMP-3 in Table 6. In these analyses, the controller model is `Llama-3-8B-Instruct`, and the environment model is `gpt-4o-mini`. As demonstrated in Table 6, `Matryoshka` enhances the summarization capabilities of the controller model. Common errors in PAG, such as inaccurate summaries or improper utilization of retrieved user history, are mitigated by `Matryoshka`, which enables the controller model to produce more accurate generations through interaction and optimization.

## G PROMPT TEMPLATES

### G.1 ALFWORLD

Following Adaplanner (Sun et al., 2024a), we implement a code-style prompt for `Matryoshka`, which can be divided into the following sections:

**High-level Planning.** The <high_level_planning> prompt is used to instruct `Matryoshka` to break the current task down into multiple subtasks, where <decompose> is replaced by a standard task decomposition process, and <receptacle_list> is substituted by the list of interactive receptacles provided by the task environment. Finally, <task> is replaced by the task description, expressed in natural language.

Table 6: Case Study for LaMP. In the "Target" column, we present the ground-truth categorization. The "Gen" column displays the final answer generated by the black-box model corresponding to each intermediate generation. The "Score" column indicates whether the generated answer ("Gen") matches the target categorization ("Target"). In the "Method" column, "Ours" refers to `Matryoshka`, while "PAG" stands for Profile Augmented Generation.

| Task | Input Question | Intermediate Generation | Method | Target | Gen | Score |
|------|----------------|-------------------------|--------|--------|-----|-------|
| 2M | Which tag does this movie relate to among the following tags? A ticking-time-bomb insomniac and a slippery soap salesman channel... | most popular tag: "dystopia", "fantasy", "comedy", "violence" | Ours | violence | violence | 1 |
| | | "Here are the most popular tags for the user: dystopia, fantasy, comedy" | PAG | violence | dystopia | 0 |
| 2N | Which category does this article relate to among the following categories? The suspect, Akayed Ullah, was the most seriously hurt in the rush-hour blast... | most popular category: politics, crime, entertainment, women, business, sports. | Ours | crime | crime | 1 |
| | | Based on the articles you provided, the most popular category written by this journalist is: poltics. | PAG | crime | politics | 0 |
| 3 | What is the score of the following review on a scale of 1 to 5? After almost 20 years in and around MIT, I've encountered only two great MIT books: (1) A.R. Gurney's out-of-print novel The Snow Ball (correction: it is Entertaining Strangers); (2) Pepper White's book... | Based on this user's past reviews, the most common positive score is: 4, with 4 reviews out of 8 receiving a score of 4. The most common negative score is: 1, with 4 reviews out of 8 receiving a score of 1. | Ours | 4 | 4 | 1 |
| | | Based on the reviews, the most common positive score is 5, and the most common negative score is 1. | PAG | 4 | 5 | 0 |

```
________ <high_level_planning> Prompt ________
# Decompose the task into steps. First give a general plan of how you
would solve the task, then for each step you plan to take, mark with
'[Step xx]'.

# Here is an example of a decomposition to the task:
# define environment
receptacles = ['diningtable 1','drawer 2', 'drawer 1', 'sinkbasin 1',
'toilet 1', 'sidetable 2', 'sidetable 1', 'cabinet 1', 'countertop 1',
'microwave 1', 'fridge 1']

<decompose>

# Here is the actual task.
# define environment
receptacles = <receptacle_list>

# <task>
# here is a decomposition:
```

**Multi-Turn Planning.**   The <multi_turn_planning> prompt is used to guide `Matryoshka` to reflect on errors in the previous decomposition and re-break the current task into multiple subtasks. Here, <predecompose> is replaced with the high-level plan from the previous turn, while all other elements retain the same meaning as before.

```
<multi_turn_planning> Prompt

# Decompose the task into steps. First give a general plan of how you
would solve the task, then for each step you plan to take, mark with
'[Step xx]'.

# Here is a successful example of a decomposition to the task:
# define environment
receptacles = ['diningtable 1','drawer 2', 'drawer 1', 'sinkbasin 1',
'toilet 1', 'sidetable 2', 'sidetable 1', 'cabinet 1', 'countertop
1', 'microwave 1', 'fridge 1']

<decompose>

# Here is the actual task.
# define environment
receptacles = <receptacle_list>

# <task>

# Here are the decomposition steps you previously generated for the
task.
<predecompose>

# However, you made a mistake in the decomposition above because of
lack of understanding of the task.

Referring to the successful example, please correct the error, if
any, and rewrite the decomposition.
# here is a decomposition:
```

**Low-level Execution.**   The <low_level_execution> prompt is used to instruct the black box model to generate a specific solution based on the problem and the plan provided by `Matryoshka`. <basic_info> defines the agent and admissible actions on Alfworld and can be found in Sun et al. (2024a). The <example> is replaced with a combination of planning and expert trajectory, while the meanings of <receptacle_list> and <task> remain consistent with the previous description. <decomposition> represents the high-level plan provided by `Matryoshka` for the current task.

```
<low_level_execution> Prompt

<basic_info>

# Now complete the function solution() below to solve the task by
composing the agent's methods to interact with the environment.
# First give a general plan of how you would solve the task, mark with
' # General Plan'. Then for each step you plan to take, 1) mark with
'[Step xx]', 2) give a reason why you think it is a good step to take
3) write an assertion to check if the step is successful.

# Here is an example of a solution to the task:
# define environment and agent
receptacles = ['diningtable 1','drawer 2', 'drawer 1', 'sinkbasin 1',
'toilet 1', 'sidetable 2', 'sidetable 1', 'cabinet 1', 'countertop 1',
'microwave 1', 'fridge 1']
agent = Agent(receptacles)

<example>

# Here is the actual task.
# define environment and agent
receptacles = <receptacle_list>
```

```
agent = Agent(receptacles)

# <task>
# here is a decomposition:
<decomposition>

# here is a solution:
```

**Planning Samples.** In ALFWorld, there are six types of tasks: `Pick`, `Clean`, `Heat`, `Cool`, `Examine`, and `Pick two`. For each type, we collect a reasonable high-level planning approach, allowing `Matryoshka` to reference them. These six planning samples are presented as follows:

Planning Sample for the task `Pick`:

<planning_sample_pick> Prompt

```
# Your task is to: put soapbar on countertop.
# here is a decomposition:
# General Plan: I need to get a list of receptacles where the soapbar
is likely to appear, and then go to each receptacle in the list until
seeing a soapbar. Then I can put get the identifier of the soapbar and
take it. Finally I can go to the countertop and put the soapbar.
# [Step 1] get a list of receptacles where the soapbar is likely to
appear.
# [Step 2] go to each receptacle in the list until seeing a soapbar.
# [Step 3] identify the soapbar I juts found and take it.
# [Step 4] go to a countertop and put the soapbar on it.
```

Planning Sample for `Clean`:

<planning_sample_clean> Prompt

```
# Your task is to: put a clean lettuce in diningtable / clean a
lettuce and put it in diningtable.
# here is a decomposition:
# General plan: I need to get a list of receptacles to find the
lettuce, take the lettuce to the sinkbasin, clean it and put it in a
diningtable.
# [Step 1] get a list of receptacles where the lettuce is likely to
appear.
# [Step 2] go to each receptacle in the list until seeing a lettuce.
# [Step 3] identify the lettuce I just found and take it.
# [Step 4] go to a sinkbasin to clean the lettuce.
# [Step 5] go to a diningtable and put the lettuce on it.
```

Planning Sample for `Heat`:

<planning_sample_heat> Prompt

```
# Your task is to: put a hot lettuce in diningtable / heat some
lettuce and put it in diningtable.
# here is a decomposition:
# General plan: I need to get a list of receptacles to find the
lettuce, take the lettuce to the microwave, heat it and put it in a
diningtable.
# [Step 1] get a list of receptacles where the lettuce is likely to
appear.
# [Step 2] go to each receptacle in the list until seeing a lettuce.
# [Step 3] identify the lettuce I juts found and take it.
# [Step 4] go to a microwave to heat the lettuce.
# [Step 5] go to a diningtable and put the lettuce on it.
```

Planning Sample for `Cool`:

<planning_sample_cool> Prompt

```
# Your task is to: put a cold lettuce in diningtable / cool some
lettuce and put it in diningtable.
```

```
# here is a decomposition:
# General plan: I need to get a list of receptacles to find the
lettuce, take the lettuce to the fridge, cool it and put it in a
diningtable.
# [Step 1] get a list of receptacles where the lettuce is likely to
appear.
# [Step 2] go to each receptacle in the list until seeing a lettuce.
# [Step 3] identify the lettuce I juts found and take it.
# [Step 4] go to a fridge to cool the lettuce.
# [Step 5] go to a diningtable and put the lettuce on it.
```

Planning Sample for `Examine`:

──────────────── `<planning_sample_examine>` Prompt ────────────────
```
# Your task is to: look at the bowl under the desklamp / examine the
bowl with the desklamp
# here is a decomposition:
# General plan: I need to get a list of receptacles to find the bowl
and take the bowl with me, then I get another list of receptacles to
find the desklamp and turn it on.
# [Step 1] get a list of receptacles where a bowl is likely to appear.
# [Step 2] go to each receptacle in the list until seeing a pen.
# [Step 3] take the bowl from the receptacle.
# [Step 4] get a list of receptacles where a desklamp is likely to
appear.
# [Step 5] go to each receptacle in the list until seeing a desklamp.
# [Step 6] turn on desklamp.
```

Planning Sample for `Pick Two`:

──────────────── `<planning_sample_picktwo>` Prompt ────────────────
```
# Your task is to: put two cellphone in cabinet / find two cellphone
and put them in cabinet
# here is a decomposition:
# General plan: I need to get a list of receptacles to find the two
cellphones, find and take the first cellphone and put it in a cabinet,
then find and take the second cellphone and put it in the cabinet.
# [Step 1] get a list of receptacles where a cellphone is likely to
appear.
# [Step 2] go to each receptacle in the list until seeing a cellphone.
# [Step 3] identify the first cellphone found and take it.
# [Step 4] go to a cabinet and put the first cellphone found on it.
# [Step 5] go to each of the remaining receptacle in the list until
seeing a second cellphone.
# [Step 6] identify the second cellphone I just found and take it.
# [Step 7] go to a cabinet and put the second cellphone found on it.
```

**Execution Samples.** Our execution sample is based on the prompt structure from Sun et al. (2024a), with the key distinction being the incorporation of the planning component. In this setup, <decompose> is substituted with the task-specific planning sample, <execution> is replaced by the expert samples from Sun et al. (2024a), and the definition of <task> remains unchanged from the previous description.

──────────────── `<execution_sample_template>` Prompt ────────────────
```
# <task>
# <decompose>
# <execution>
```

**Close-loop Refinement.** To implement close-loop refinement during the inference stage, we follow the approach from Sun et al. (2024a) and introduce several prompts: a <code_check> prompt to identify and fix any syntax errors during execution generation, a <refinement> prompt to address refinement in case of assertion errors, and a <start_from> prompt to determine the starting point for

the new solution after revising the plan. Detailed descriptions of these prompts can be found in Sun et al. (2024a).

### G.2 GSM-HARD

Following PAL framework (Gao et al., 2023), we implement a code-based framework to solve mathematical problems on GSM-Hard, which is primarily divided into two steps: `Matryoshka` breaks down the mathematical problem into sub-problems, and the black-box model converts each sub-problem into a code block.

**Problem Decomposition.** For `Matryoshka`, we employ a three-shot prompt to guide the decomposition steps, where <question> represents the current problem.

```
                     <problem_decomposition> Prompt
# System Message: You will decompose a math problem into smaller
parts. Follow the prompt instruction and do not generate redundant
information.

Q: Olivia has $23. She bought five bagels for $3 each. How much money
does she have left?
A: Let's break down this problem:\nHow much does Olivia spend on
bagels?\nHow much money does Olivia have left after the purchase?

Q: Michael had 58 golf balls. On tuesday, he lost 23 golf balls. On
wednesday, he lost 2 more. How many golf balls did he have at the end
of wednesday?
A: Let's break down this problem:\nHow many golf balls did Michael
lose in total by the end of Wednesday?\nHow many golf balls does
Michael have left after losing the total amount?

Q: There were nine computers in the server room. Five more computers
were installed each day, from monday to thursday. How many computers
are now in the server room?
A: Let's break down this problem:\nHow many computers were added in
total from Monday to Thursday?\nHow many computers are now in the
server room after adding the new ones?

Q: <question>
A:
```

**Multi-turn Decomposition.** If the Black-box LLM encounters an error while solving the problem, `Matryoshka` is required to reflect on the previous decomposition <decompose> and re-decompose the problem.

```
                   <multi_turn_decomposition> Prompt
# System Message: You will decompose a math problem into smaller
parts. Follow the prompt instruction and do not generate redundant
information.

# Here are some examples on how to decompose the question into smaller
parts.
Q: Olivia has $23. She bought five bagels for $3 each. How much money
does she have left?
A: Let's break down this problem:\nHow much does Olivia spend on
bagels?\nHow much money does Olivia have left after the purchase?

Q: Michael had 58 golf balls. On tuesday, he lost 23 golf balls. On
wednesday, he lost 2 more. How many golf balls did he have at the end
of wednesday?
A: Let's break down this problem:\nHow many golf balls did Michael
lose in total by the end of Wednesday?\nHow many golf balls does
Michael have left after losing the total amount?
```

```
Q: There were nine computers in the server room. Five more computers
were installed each day, from monday to thursday. How many computers
are now in the server room?
A: Let's break down this problem:\nHow many computers were added in
total from Monday to Thursday?\nHow many computers are now in the
server room after adding the new ones?

# Here is the actual question.
Q: <question>
You have decomposed the problem into smaller parts:
<decompose>
However, you made a mistake in the decomposition above because of lack
of understanding of the question.

Referring to the examples, please correct the error, if any, and
rewrite the decomposition.
A:
```

**Code Generation.** Given the problem and the decomposition provided by `Matryoshka`, the Black-box model generates the corresponding code block for each sub-problem. We continue to use a three-shot prompt to instruct the Black-box model on how to translate the sub-problems into code, where <question> represents the current problem and <decompose> represents the decomposition provided by `Matryoshka`.

```
───────────────────────── <code_generation> Prompt ─────────────────────────
# System Message: You will write python program to solve math
problems. You will write annotations and code blocks following
instructions. Annotations should be written in the form of a question.

Let's use python to solve math problems. Here are three examples how
to do it,
Q: Olivia has $23. She bought five bagels for $3 each. How much money
does she have left?
Let's break down this problem:\nHow much does Olivia spend on bagels?
\nHow much money does Olivia have left after the purchase?
```
def solution():
    """Olivia has $23. She bought five bagels for
    $3 each. How much money does she have left?"""
    # Initialization of variables
    money_initial = 23
    bagels = 5
    bagel_cost = 3

    # How much does Olivia spend on bagels?
    money_spent = bagels * bagel_cost

    # How much money does Olivia have left after the purchase?
    money_left = money_initial - money_spent
    result = money_left
    return result
```

Q: Michael had 58 golf balls. On tuesday, he lost 23 golf balls. On
wednesday, he lost 2 more. How many golf balls did he have at the end
of wednesday?
Let's break down this problem:\nHow many golf balls did Michael lose
in total by the end of Wednesday?\nHow many golf balls does Michael
have left after losing the total amount?
```
def solution():
    """Michael had 58 golf balls. On tuesday, he lost 23 golf balls.
    On wednesday, he lost 2 more. How many golf balls did he have at
    the end of wednesday?"""
```

```
    # Initialization of variables
    golf_balls_initial = 58
    golf_balls_lost_tuesday = 23
    golf_balls_lost_wednesday = 2

    # How many golf balls did Michael lose in total by the end of
    Wednesday?
    golf_balls_left = golf_balls_initial - golf_balls_lost_tuesday -
    golf_balls_lost_wednesday

    # How many golf balls does Michael have left after losing the
    total amount?
    result = golf_balls_left
    return result
```

Q: There were nine computers in the server room. Five more computers
were installed each day, from monday to thursday. How many computers
are now in the server room?
Let's break down this problem:\nHow many computers were added in total
from Monday to Thursday?\nHow many computers are now in the server
room after adding the new ones?
```
def solution():
    """There were nine computers in the server room. Five more
    computers were installed each day, from monday to thursday. How
    many computers are now in the server room?"""
    # Initialization of variables
    computers_initial = 9
    computers_per_day = 5
    num_days = 4  # 4 days between monday and thursday

    # How many computers were added in total from Monday to Thursday?
    computers_added = computers_per_day * num_days

    # How many computers are now in the server room after adding the
    new ones?
    computers_total = computers_initial + computers_added
    result = computers_total
    return result
```

How about this question?
Q: <question>
<decompose>

**Close-loop Refinement.** <refinement> prompt is employed to encourage the model to reflect and fix issues in its own solution, wherein <error_msg> is replaced by the error message returned by the solution function.

---
<refinement> Prompt
---
```
Let's use python to solve math problems. Here are three successful
cases on how to do it,
Q: Olivia has $23. She bought five bagels for $3 each. How much money
does she have left?
```
def solution():
    """Olivia has $23. She bought five bagels for $3 each. How much
    money does she have left?"""
    money_initial = 23
    bagels = 5
    bagel_cost = 3
    money_spent = bagels * bagel_cost
    money_left = money_initial - money_spent
```

```
      result = money_left
      return result
```

Q: Michael had 58 golf balls. On tuesday, he lost 23 golf balls. On
wednesday, he lost 2 more. How many golf balls did he have at the end
of wednesday?
```
def solution():
    """Michael had 58 golf balls. On tuesday, he lost 23 golf balls.
    On wednesday, he lost 2 more. How many golf balls did he have at
    the end of wednesday?"""
    golf_balls_initial = 58
    golf_balls_lost_tuesday = 23
    golf_balls_lost_wednesday = 2
    golf_balls_left = golf_balls_initial - golf_balls_lost_tuesday -
    golf_balls_lost_wednesday
    result = golf_balls_left
    return result
```

Q: There were nine computers in the server room. Five more computers
were installed each day, from monday to thursday. How many computers
are now in the server room?
```
def solution():
    """There were nine computers in the server room. Five more
    computers were installed each day, from monday to thursday. How
    many computers are now in the server room?"""
    computers_initial = 9
    computers_per_day = 5
    num_days = 4  # 4 days between monday and thursday
    computers_added = computers_per_day * num_days
    computers_total = computers_initial + computers_added
    result = computers_total
    return result
```

# Here is the actual question.
Q: <question>
You have generated code of solution() to solve the task. However, you
executed the solution() function and get an error message:
<error_msg>

Referring to the successful case and the error message, you should
complete the solution function with the correct code.

### G.3  LAMP

Following the RAG-based framework (Salemi et al., 2023) and the PAG-based framework (Richardson et al., 2023), we implement prompt designs for both Matryoshka and the baseline methods. The prompt design for RAG is presented in Table 7, while the prompts for the two-stage PAG and Matryoshka are shown in Table 8. We create prompts for the controller model using the templates from Table 8 and subsequently combine the intermediate generations with the input question to form prompts for the environment model. Since LaMP-3 prompts are particularly lengthy, we provide additional examples of our PAG prompts for LaMP-1, LaMP-2N, LaMP-2M, and LaMP-4 as follows.

Table 7: RAG prompt design for five LaMP tasks. Concat(·) concatenates the input strings in order, and PPEP(·) composes the prompt for each retrieved item from the profile. [INPUT] represents the task's input.

| Task | Per Profile Entry Prompt (PPEP) | Aggregated Input Prompt (AIP) |
|------|-------------------------------|------------------------------|
| LaMP-1 | "$P_i$[title]" | concat([PPEP($P_1$), ..., PPEP($P_n$)], ", and "). [INPUT] |
| LaMP-2N | "the category for the article: "$P_i$[text]" is ""$P_i$[category]"" | concat([PPEP($P_1$), ..., PPEP($P_n$)], ", and "). [INPUT] |
| LaMP-2M | "the tag for the movie: "$P_i$[description]" is "$P_i$[tag]" | concat([PPEP($P_1$), ..., PPEP($P_n$)], ", and "). [INPUT] |
| LaMP-3 | $P_i$[score] is the score for "$P_i$[text]" | concat([PPEP($P_1$), ..., PPEP($P_n$)], ", and "). [INPUT] |
| LaMP-4 | "$P_i$[title]" is the title for "$P_i$[text]" | concat([PPEP($P_1$), ..., PPEP($P_n$)], ", and "). [INPUT] |

Table 8: Summarization prompt design for the five LaMP tasks. [INPUT] represents the task's input.

| Task | Prompt |
|------|--------|
| LaMP-1 | Write a summary, in English, of the research interests and topics of a researcher who has published the following papers. Only generate the summary, no other text. |
| LaMP-2N | Look at the following past articles this journalist has written and determine the most popular category they write in. Answer in the following format: most popular category: <category top1>, <category top2>, ..., <category topn> |
| LaMP-2M | Look at the following past movies this user has watched and determine the mostpopular tag they labeled. Answer in the following form: most popular tag: <tag top1>, <tag top2>, ..., <tag topn> |
| LaMP-3 | Based on this user's past reviews, what are the most common scores they give for positive and negative reviews? Answer in the following form: most common positive score: <most common positive score>, most common negative score: <most common negative score> |
| LaMP-4 | Given this author's previous articles, try to describe a template for their headlines. I want to be able to accurately predict the headline gives one of their articles. Be specific about their style and wording, don't tell me anything generic. Use the following format: The template is: '[template 1]', '[template 2]', '[template 3]', '[template 4]' |

```
┌──────────────── PAG Prompt Demo for LaMP-1 ────────────────┐
Write a summary, in English, of the research interests and topics
of a researcher who has published the following papers.
Only generate the summary, no other text.
The published papers are:
    \"Efficient Evaluation of Continuous Text Search Queries\",
    and \"Continuous Monitoring of Spatial Queries in Wireless
    Broadcast Environments\", and \"Spatial queries in wireless
    broadcast environments\", and \"Maximum Rank Query\", and
    \"Anonymous Query Processing in Road Networks\",
    and \"An Incremental Threshold Method for Continuous Text
    Search Queries\", and \"Continuous Top-k Monitoring on
    Document Streams.\", and \"Best upgrade plans for large road
    networks\", and \"Scalable verification for outsourced dynamic
    databases\", and \"Heuristic algorithms for balanced multi-way
    number partitioning\", and \"Aggregate nearest neighbor
    queries in spatial databases\", and \"Partially materialized
    digest scheme: an efficient verification method for outsourced
    databases\", and \"Best upgrade plans for single and multiple
    source-destination pairs.\", and \"Tree-based partition
    querying: a methodology for computing medoids in large spatial
    datasets\", and \"A Threshold-Based Algorithm for Continuous
    Monitoring of k Nearest Neighbors\", and \"Computing immutable
    regions for subspace top-k queries\", and \"Historical traffic-
    tolerant paths in road networks\"...
```

```
┌─────────── PAG Prompt Demo for LaMP-2M ───────────┐
Look at the following past movies this user has watched and determine
the most popular tag they labeled. Answer in the following form:
most popular tag: <tag top1>, <tag top2>, ..., <tag topn>.
The movies and tags are:
    the tag for the movie: \"Young hobbit Frodo Baggins,
    after inheriting a mysterious ring from his uncle Bilbo,
    must leave his home in order to keep it from falling into
    the hands of its evil creator. Along the way,
    a fellowship is formed to protect the ringbearer
    and make sure that the ring arrives at its final destination:
    Mt. Doom, the only place where it can be destroyed.\" is
    \"fantasy\" , and the tag for the movie: \"Set in the 22nd
    century, The Matrix tells the story of a computer hacker
    who joins a group of underground insurgents fighting the vast
    and powerful computers who now rule the earth.\" is \"sci-fi\" ,
    and the tag for the movie: \"Batman raises the stakes in his
    war on crime. With the help of Lt. Jim Gordon and District
    Attorney Harvey Dent, Batman sets out to dismantle the
    remaining criminal organizations that plague the streets. The
    partnership proves to be effective, but they soon find
    themselves prey to a reign of chaos unleashed by a rising
    criminal mastermind known to the terrified citizens of Gotham
    as the Joker.\" is \"psychology\" , and the tag for the movie:
    \"An unsuspecting, disenchanted man finds himself working as a
    spy in the dangerous, high-stakes world of corporate
    espionage. Quickly getting way over-his-head, he teams up with
    a mysterious femme fatale.\" is \"twist ending\"...
└───────────────────────────────────────────────────┘
```

```
┌─────────── PAG Prompt Demo for LaMP-2N ───────────┐
Look at the following past articles this journalist has
written and determine the most popular category they write in.
Answer in the following format: most popular category:
<category top1>, <category top2>, ..., <category topn>.
The articles and categories are:
    the category for the article:
    \"Champions like Tiger Woods are always charting and changing
    their course to be certain everything is on track. Tiger
    didn't just come to Augusta because it was the popular thing
    to do. He wouldn't have showed up if he wasn't ready to win.
    He came to win and he's prepared to win.\" is \"sports\" , and
    the category for the article: \"In 2011, in an interview with
    The Golf Channel, I predicted a Tiger Woods comeback while
    many others said he was done. I was right that time and I am
    right again, and I'll say it right now and on the record:
    Tiger Woods will be back again and dominate the game of golf
    like the Tiger of old.\" is \"sports\" , and the category for
    the article: \"What do you teach your kids about money,
    prosperity and how to get rich? If you\u2019re like most
    parents, the answer is probably\" is \"business\" , and the
    category for the article: \"With a little bit of planning and
    a lot of discipline, accomplishing your goals in the New Year
    can become a reality.  Imagine the immense satisfaction you'll
    feel at this same time next year when you can look back and
    look at how far you've come and all that you have
    accomplished.\" is \"healthy living\" , and the category for
    the article: \"This whole argument boils down to a simple
    premise: who is in charge of our lives? Doctors? Politicians?
    Religious leaders? Or Us? Are we so feeble minded that we
    cannot be trusted to be responsible for our own existence?\"
    is \"politics\"...
└───────────────────────────────────────────────────┘
```

```
┌─────────────── PAG Prompt Demo for LaMP-4 ───────────────┐
│ Given this author's previous articles, try to describe a  │
│ template for their headlines. I want to be able to accurately│
│ predict the headline gives one of their articles. Be specific│
│ about their style and wording, don't tell me anything generic.│
│ Use the following format: The template is: '[template 1]', │
│ '[template 2]', '[template 3]', '[template 4]'.          │
│ Previous articles and titles are:                         │
│     \"Selling a House to Buy a House\" is the title for \"Homeowners│
│     sell their homes and buy other homes for a variety    │
│     of reasons including a need to live closer            │
│     to a place of employment, to be closer to family, to enjoy a│
│     better climate, or simply to upgrade. This article is about│
│     finding the best sequence of steps in the process.\", and│
│     \"Investing In a Larger Down Payment: High Yields and No│
│     Risk\" is the title for \"Consumers looking to purchase a home│
│     within the near future face many decisions, including how│
│     large a down payment to make. The down payment is the sale│
│     price (confirmed by a appraisal) less the loan amount. In most│
│     cases, home purchasers must have financial assets at least as│
│     large as the down payment they make.\", and \"Why and How to│
│     Eliminate Mortgage Charges by Third Parties\" is the title for│
│     \"Third-party settlement costs could be eliminated by  │
│     implementation of one simple rule: any service required by│
│     lenders as a condition for the granting of a home mortgage│
│     must be purchased and paid for by the lender.\", and \"Do Home│
│     Buyers Need a Pre-Approval?\" is the title for \"With  │
│     bargaining power shifting from home buyers to sellers in an│
│     increasing number of local markets, buyers in competition with│
│     other buyers are looking for any edge they can get. One│
│     possible edge is a pre-approval letter (henceforth PAL) from a│
│     lender.\", and \"A New Challenge to the HECM Reverse Mortgage│
│     Program\" is the title for \"The United States today faces a│
│     retirement funds crisis: a rapidly growing number of persons│
│     who are retiring without the financial capacity to support│
│     themselves during ever-increasing life spans.\"...      │
└──────────────────────────────────────────────────────────┘
```

