# OpenReview forum: "Martryoshka: Learning to Drive Black-Box LLMs with LLMs"
_ICLR.cc/2025/Conference — Submitted to ICLR 2025_

### Official Review · Reviewer_zosj · 2024-10-27

**Soundness:** 2
**Presentation:** 2
**Contribution:** 2
**Rating:** 5
**Confidence:** 4

**Summary:**

This paper presents a novel multi-turn framework where a white-box LLM (e.g., Llama) generates prompts, and a black-box LLM (e.g., GPT-4) responds to each prompt in turn. It proposes a DPO-based algorithm to train such white-box LLM. It demonstrates superior performance in various personalization, reasoning, and planning tasks.

**Strengths:**

- The paper is well-written and easy to follow. Specifically, Figure 1 alone clearly illustrates the paper’s goal.
- The paper evaluates the proposed method across three different tasks, demonstrating its broad applicability.
- The plug-and-play experiment is impressive, showing that training a white-box LLM specifically for a particular black-box LLM is unnecessary, thereby demonstrating its generalizability.

**Weaknesses:**

- The proposed method definitely echoes black-box prompt optimization, as mentioned by the author in the related work section. The only difference is single-turn versus multi-turn interaction. Also, there is some recent work on black-box prompt optimization for non-classification tasks [1].
- I believe that the naive DPO is not suitable for multi-turn MDPs when the transition dynamics are non-deterministic [2]. The authors should justify why they use DPO instead of REINFORCE or PPO.
- The proposed method uses more capable LLMs (e.g., GPT-4) for training, which makes the comparison unfair. The authors should provide results for the proposed method using the same LLM as the baseline models.

[1] Jiale Cheng et al., Black-Box Prompt Optimization: Aligning Large Language Models without Model Training, ACL 2024 \
[2] Wentao Shi et al., Building Math Agents with Multi-Turn Iterative Preference Learning, arXiv 2024

**Questions:**

See the weaknesses above.

---

> ### Author Response · Authors · 2024-11-22
>
> Thank you for your detailed suggestions and comments. Please find the corresponding responses as follows:
>
> **[Weakness 1]**
>
> Thank you for the suggestion. First, it is crucial to underscore the fundamental difference between our approach and conventional black-box prompt optimization. Rather than optimizing prompts specifically for individual tasks, our focus is on developing a **generalized model** that can generate prompts applicable to a wide range of tasks. In traditional black-box prompt optimization, fine-tuning an instance often results in limited transferability to other scenarios. In contrast, our objective is to deliver a more **versatile and adaptable personal assistant** capable of supporting users across various contexts and applications.
>
> In [1], BPO also employs a RL method to train a prompt optimization model, aligning the outputs of black-box LLMs more closely with human preferences across multiple instruction datasets by optimizing the input prompts.
>
> The key differences are as follows:
>
> 1. BPO's training data is derived by filtering open-sourced feedback data, whereas our approach uses outputs directly generated by a white-box LLM, offering broader applicability.
> 2. BPO trains using single-turn interactions, while our method incorporates multi-turn interactions, allowing for a more comprehensive optimization process.
>
> Thanks again for your insightful suggestion, and we will add this meaningful and highly relevant discussion and citation into our paper.
>
> Reference:
>
> [1] Jiale Cheng et al., Black-Box Prompt Optimization: Aligning Large Language Models without Model Training, ACL 2024

---

> > ### Author Response · Authors · 2024-11-22
> >
> > **[Weakness 2]**
> >
> > We appreciate the suggestion and would like to elaborate on the reinforcement learning method we chose. First, we improved upon the basic DPO method by introducing an iterative guidance optimization approach that interleaves data sampling with model training steps. Through extensive experiments, we demonstrated that this method—which incorporates multi-turn interactions with the environment—significantly outperforms simple DPO training.
> >
> > Furthermore, it is important to emphasize that the core contribution of our work lies in adapting the pioneering controller-generator framework. This framework is highly flexible and enables controllable multi-turn generation, significantly enhancing the ability of black-box large language models to handle complex, long-horizon tasks.
> >
> > The specific RL method used to train the controller model is simply a design choice and not the primary focus of our work. Nevertheless, to showcase the adaptability of our framework, we conducted experiments on all three datasets using SimPo [1] as the RL method. The results, presented below, demonstrate that our framework consistently delivers strong performance across different RL methods, further validating its effectiveness and generality.
> >
> > **AlfWorld**
> >
> > | Methods(↓) Tasks(→)       | Pick       | Clean     | Heat       | Cool       | Examine   | Pick Two  | All       |
> > | ------------------------- | ---------- | --------- | ---------- | ---------- | --------- | --------- | --------- |
> > | Adaplanner                | 100.00     | 93.55     | 78.26      | 95.24      | 66.67     | 88.24     | 88.06     |
> > | Matry+simpo               | **100.00** | **93.55** | **95.65**  | **95.24**  | **77.78** | **88.24** | **92.54** |
> > | w/o $1^{st}$-round SimPO  | 100.00     | 93.55     | 86.96      | 95.24      | 55.56     | 88.24     | 88.06     |
> > | w/o Guidance Optimization | 100.00     | 93.55     | 91.30      | 85.71      | 11.11     | 88.24     | 81.34     |
> > | Matry + dpo               | **100.00** | **93.55** | **100.00** | **100.00** | **83.33** | **88.24** | **94.78** |
> >
> >
> >
> > **GSM**
> >
> > | Dataset(→) Methods(↓) | GSM8K(gpt-3.5) | GSM8K(4o-mini) | GSM-HARD(gpt-3.5) | GSM-HARD(4o-mini) |
> > | --------------------- | -------------- | -------------- | ----------------- | ----------------- |
> > | CoT                   | 0.809          | 0.932          | 0.406             | 0.5               |
> > | Least-to-Most         | 0.811          | 0.908          | 0.425             | 0.498             |
> > | PAL                   | 0.802          | 0.920          | 0.638             | 0.748             |
> > | PAL_{Self-Debug}      | 0.864          | 0.943          | 0.701             | 0.774             |
> > | Matry+simpo           | **0.908**      | **0.950**      | **0.731**         | **0.789**         |
> > | Matry+dpo             | **0.911**      | **0.956**      | **0.738**         | **0.779**         |
> >
> >
> >
> > **LaMP**
> >
> > |                 | Acc/F-1 ⬆️       | Acc/F-1 ⬆️       | Acc/F-1 ⬆️       | mae/rmse ⬇️      | R-1/R-L/BLEU ⬆️        |
> > | --------------- | --------------- | --------------- | --------------- | --------------- | --------------------- |
> > |                 | LaMP1           | LaMP-2N         | LaMP-2M         | LaMP-3          | LaMP-4 prompt1        |
> > | llama3 + 4omini | 0.611/0.611     | 0.807/0.575     | 0.496/0.432     | 0.311/0.636     | 0.131/0.120/1.341     |
> > | 4omini + 4omini | 0.624/0.624     | 0.775/0.559     | 0.496/0.443     | 0.316/0.645     | 0.143/0.130/1.968     |
> > | Matry+dpo   | **0.640/0.640** | **0.832/0.614** | **0.535/0.475** | **0.282/0.588** | **0.171/0.157/4.144**     |
> > | Matry+simpo | **0.628/0.628**     | **0.826/0.598**     | **0.522/0.461**     | **0.294/0.614**     | **0.180/0.167/4.997** |
> >
> > Reference:
> >
> > [1] Meng, Y., Xia, M. and Chen, D., 2024. Simpo: Simple preference optimization with a reference-free reward. *arXiv preprint arXiv:2405.14734*.

---

> > > ### Author Response · Authors · 2024-11-22
> > >
> > > **[Weakness 3]**
> > >
> > > Thank you for the suggestion. In practice, we only utilized the GPT-4o model in collecting DPO training set for the LaMP dataset. This was because the LaMP dataset lacks ground truth annotations, and in cases where the original black-box model could not generate sufficient positive data, we leveraged the GPT-4o model to produce additional positive samples needed for DPO training. Therefore, the use of the GPT-4o model was simply a design choice for data annotation.
> > >
> > > If alternative options were available, such as high-quality human-annotated data, ground truth chain-of-thoughts from GSM8K, or interaction-based feedback from Alfworld to determine the quality of data, we could entirely bypass the use of the GPT-4o model.
> > >
> > > In fact, even without the GPT-4o model, we can still achieve sufficiently or better positive results on the LaMP dataset. As shown in Table A, although slightly lower than the metrics reported using GPT-4o for LaMP-1 and LaMP-2, the results get even better on LaMP-3 and LaMP-4 and remain significantly better than other baselines on all LaMP tasks. As for the fraction, we observe that less than 5% data is from gpt-4o, further claiming that this setting is not needed. This demonstrates that our approach works well even without the assistance of a powerful model. In this case, where the powerful model is removed, it further highlights that the improvement in experimental performance stems from **self-improvement** rather than being directly distilled from a powerful model like GPT-4o.
> > >
> > > **Table A: Experimental results on the LaMP benchmark without GPT-4o-based labeling**
> > >
> > > |                                 | LaMP-1(ACC/F-1) | LaMP-2N(ACC/F-1) | LaMP-2M(ACC/F-1) | LaMP-3⬇️(MAE/RMSE) | LaMP-4(R-1/R-L/BLEU) |
> > > | ------------------------------- | ------------------- | -------------------- | -------------------- | --------------------- | ------------------------ |
> > > | LLaMA3-8B-Instruct              | 0.611/0.611         | 0.807/0.575          | 0.496/0.432          | 0.311/0.636           | 0.131/0.120/1.341        |
> > > | gpt-4o-mini                     | 0.624/0.624         | 0.775/0.559          | 0.496/0.443          | 0.316/0.645           | 0.143/0.130/1.968        |
> > > | gpt-4o-mini(zero-shot)          | 0.514/0.513         | 0.655/0.473          | 0.413/0.325          | 0.371/0.673           | 0.132/0.116/0.992        |
> > > | Martryoshka with 4o-Labeling    | **0.640/0.640**     | **0.832/0.614**      | **0.535/0.475**      | **0.282/0.588**       | **0.171/0.157/4.144**    |
> > > | Martryoshka without 4o-Labeling | **0.640/0.639**     | **0.823/0.607**      | **0.527/0.465**      | **0.277/0.581**       | **0.174/0.160/4.298**    |
> > >
> > > Additionally, we present experimental results below where the controller model is replaced with more capable black-box models across all three tasks. The results demonstrate that our method still achieves better performance across various tasks, further proving the effectiveness of our approach.
> > >
> > > **GSM**
> > >
> > > |                         | GSM8K     | GSM-Hard  |
> > > | ----------------------- | --------- | --------- |
> > > | gpt3.5 + gpt3.5         | 0.896     | 0.734     |
> > > | Matryoshka + gpt3.5     | **0.911** | **0.738** |
> > > | gpt4o-mini + gpt4o-mini | 0.948     | **0.791** |
> > > | Matryoshka + gpt4o-mini | **0.956** | 0.779     |
> > >
> > > **Alfworld:**
> > >
> > > | **Methods**          | **Pick**   | **Clean** | **Heat**   | **Cool**  | **Examine** | **Pick Two** | **All(134 tasks)** |
> > > | -------------------- | ---------- | --------- | ---------- | --------- | ----------- | ------------ | ------------------ |
> > > | gpt3.5 + gpt3.5      | 100.00     | 41.94     | 100.00     | 76.19     | 88.89       | 88.24        | 79.85              |
> > > | gpt4o-mini + gpt3.5  | 95.83      | 45.16     | 56.52      | 52.38     | 5.56        | 88.24        | 57.46              |
> > > | Martryoshka + gpt3.5 | **100.00** | **93.55** | **100.00** | **90.48** | **100.00**  | **88.24**    | **95.52**          |
> > >
> > > **LaMP:**
> > >
> > > |                         | Acc/F-1 ⬆️       | Acc/F-1 ⬆️       | Acc/F-1 ⬆️       | mae/rmse ⬇️      | R-1/R-L/BLEU ⬆️        |
> > > | ----------------------- | --------------- | --------------- | --------------- | --------------- | --------------------- |
> > > |                         | LaMP1           | LaMP-2N         | LaMP-2M         | LaMP-3          | LaMP-4 prompt1        |
> > > | gpt4o-mini + gpt4o-mini | 0.624/0.624     | 0.775/0.559     | 0.496/0.443     | 0.316/0.645     | 0.143/0.130/1.968     |
> > > | Matryoshka + gpt4o-mini | **0.640/0.640** | **0.832/0.614** | **0.535/0.475** | **0.282/0.588** | **0.171/0.157/4.144** |
> > > | Martryoshka + gpt3.5    | 0.594/0.593     | 0.798/0.609     | 0.469/0.412     | 0.286/0.599     | 0.176/0.161/4.222     |
> > > | gpt3.5 + gpt3.5         | 0.590/0.589     | 0.790/0.594     | 0.399/0.325     | 0.357/0.693     | 0.166/0.150/3.433     |

---

> > > > ### Author Response · Authors · 2024-11-25
> > > > **Gentle reminder: The interactive discussion period will end in less than two days**
> > > >
> > > > Dear Reviewer zosj,
> > > >
> > > > Thank you again for your time and efforts in reviewing our paper.
> > > >
> > > > As the discussion period draws close, we kindly remind you that two days remain for further comments or questions. We would appreciate the opportunity to address any additional concerns you may have before the discussion phase ends.
> > > >
> > > > Thank you very much.
> > > >
> > > > Best regards,
> > > >
> > > > Authors

---

> > > > > ### Author Response · Authors · 2024-12-01
> > > > > **Gentle reminder: The author/reviewer discussion phase will conclude on December 2nd.**
> > > > >
> > > > > Dear Reviewer zosj,
> > > > >
> > > > > We sincerely appreciate your valuable and constructive feedback. We would like to kindly remind you that the author/reviewer discussion phase concludes on December 2nd. We hope our responses and additional experiments have addressed your concerns and improved the paper’s quality. If you have any further suggestions or comments, please feel free to share them. We are looking forward to a constructive discussion during the rebuttal phase.
> > > > >
> > > > > Best Regards,
> > > > >
> > > > > The Authors

---

### Official Review · Reviewer_Li83 · 2024-11-03

**Soundness:** 3
**Presentation:** 3
**Contribution:** 2
**Rating:** 6
**Confidence:** 3

**Summary:**

This paper fine-tunes a white-box model, like Llama, to generate improved prompts—such as question analysis—for black-box models like GPT. The approach combines supervised behavior cloning and iterative guidance optimization, achieving better performance than using the original white-box and black-box models directly.

**Strengths:**

1. The writing is clear and straightforward.
2. They address various tasks, including personalization, math, and planning, and design specific workflows for each.

**Weaknesses:**

I'm confused about the necessity of using a white-box model as a controller. This setup follows a standard agent workflow: the first agent performs analysis and decomposes the task into steps, while the second agent generates the final answer. This is a common approach in agent research, offering no much novelty. Here, they replace the first agent with a white-box model (Llama) and apply SFT and RL to improve it. However, during SFT, they use GPT-4 or GPT-3.5 to generate training data for Llama, so a more appropriate baseline comparison would be GPT-4 + GPT-4 rather than Llama + GPT-4. I also question if Llama-SFT-IGO + GPT-4 actually outperforms GPT-4 + GPT-4, especially since current results show only marginal improvement of Llama-SFT-IGO + GPT-4 over Llama + GPT-4.

**Questions:**

What's the result of directly using two black-box models directly, rather than first distilling the black-box models onto white-box models and then applying RL?

---

> ### Author Response · Authors · 2024-11-22
>
> We would like to thank the Reviewer for the insightful feedback. Your suggestions for clarification and improvement are constructive. Please find the responses below:
>
> **[Weakness]**
>
> Thank you for the suggestion. From a motivational perspective, our primary goal is to enable model personalization. While existing black-box LLMs excel across various tasks, they lack the ability to be trained for personalized usage. Through our proposed controller-generator framework, we can train a personalized white-box LLM to guide the black-box LLM, enabling personalized interactions and usage.
>
> Additionally, we selected the LLaMA3-8B-Instruct model as the controller because it is significantly more lightweight compared to black-box LLMs like GPT. Our research shows that even a lightweight model can achieve state-of-the-art performance across diverse tasks, including reasoning, planning, and personalization, effectively balancing high performance with practical convenience.
>
> **[Questions]**
>
> It is also interesting to compare the results of directly using two black-box models with those achieved by our controller-generator framework. In the following sections, we will first present the superior performance of our approach compared to the two black-box models across various tasks, followed by an overall analysis of the results.
>
> **GSM**
>
> |                         | GSM8K     | GSM-Hard  |
> | ----------------------- | --------- | --------- |
> | gpt3.5 + gpt3.5         | 0.896     | 0.734     |
> | Matryoshka + gpt3.5     | **0.911** | **0.738** |
> | gpt4o-mini + gpt4o-mini | 0.948     | **0.791** |
> | Matryoshka + gpt4o-mini | **0.956** | 0.779     |
>
> **Alfworld:**
>
> | **Methods**          | **Pick**   | **Clean** | **Heat**   | **Cool**  | **Examine** | **Pick Two** | **All(134 tasks)** |
> | -------------------- | ---------- | --------- | ---------- | --------- | ----------- | ------------ | ------------------ |
> | gpt3.5 + gpt3.5      | 100.00     | 41.94     | 100.00     | 76.19     | 88.89       | 88.24        | 79.85              |
> | gpt4o-mini + gpt3.5  | 95.83      | 45.16     | 56.52      | 52.38     | 5.56        | 88.24        | 57.46              |
> | Martryoshka + gpt3.5 | **100.00** | **93.55** | **100.00** | **90.48** | **100.00**  | **88.24**    | **95.52**          |
>
> **LaMP:**
>
> |                         | Acc/F-1 ⬆️       | Acc/F-1 ⬆️       | Acc/F-1 ⬆️       | mae/rmse ⬇️      | R-1/R-L/BLEU ⬆️        |
> | ----------------------- | --------------- | --------------- | --------------- | --------------- | --------------------- |
> |                         | LaMP1           | LaMP-2N         | LaMP-2M         | LaMP-3          | LaMP-4 prompt1        |
> | gpt4o-mini + gpt4o-mini | 0.624/0.624     | 0.775/0.559     | 0.496/0.443     | 0.316/0.645     | 0.143/0.130/1.968     |
> | Matryoshka + gpt4o-mini | **0.640/0.640** | **0.832/0.614** | **0.535/0.475** | **0.282/0.588** | **0.171/0.157/4.144** |
> | Martryoshka + gpt3.5    | 0.594/0.593     | 0.798/0.609     | 0.469/0.412     | 0.286/0.599     | 0.176/0.161/4.222     |
> | gpt3.5 + gpt3.5         | 0.590/0.589     | 0.790/0.594     | 0.399/0.325     | 0.357/0.693     | 0.166/0.150/3.433     |
>
> **Note**: For tasks in ALFWorld, we adhered to the setup used in Adaplanner [1]. Since the prompt instructions are highly dependent on the type of black-box LLM, certain tasks cannot be solved when using GPT-4o-mini as the black-box LLM. Therefore, we standardized GPT-3.5-turbo as the black-box LLM for these experiments.
>
> As shown, compared to different two-black-box configurations (e.g., GPT-3.5-turbo-0125 and GPT-4o-mini), our framework consistently achieves either comparable or significantly better results in most cases. We attribute this to the following key reason:
>
> **Effective Problem Decomposition with Feedback**: As highlighted in our abstract, we treat the black-box LLM within our framework as an "environment" and the white-box LLM as a "controller." The white-box LLM decomposes the problem and inputs it to the black-box LLM. The black-box LLM interacts with the environment or compares its output against ground truth, returning the feedback as a supervisory signal. This feedback helps filter high-quality problem decompositions to train the white-box LLM. Consequently, the trained white-box LLM generates problem decompositions that more effectively guide the black-box LLM to solve tasks. In contrast, a black-box LLM alone lacks environment feedback and cannot perform equally effective problem decomposition.
>
> Reference:
>
> [1] Sun, Haotian, Yuchen Zhuang, Lingkai Kong, Bo Dai, and Chao Zhang. "Adaplanner: Adaptive planning from feedback with language models." *Advances in Neural Information Processing Systems* 36 (2024).

---

> > ### Author Response · Authors · 2024-11-25
> > **Gentle reminder: The interactive discussion period will end in less than two days**
> >
> > Dear Reviewer Li83,
> >
> > Thank you again for your time and efforts in reviewing our paper.
> >
> > As the discussion period draws close, we kindly remind you that two days remain for further comments or questions. We would appreciate the opportunity to address any additional concerns you may have before the discussion phase ends.
> >
> > Thank you very much.
> >
> > Best regards,
> >
> > Authors

---

> > > ### Comment · Reviewer_Li83 · 2024-11-27
> > >
> > > Thank you to the authors for their response. The proposed method appears to demonstrate some improvement over stronger baselines. I increase my score to 6 and hope the method can show more strength in other more suitable settings.

---

### Official Review · Reviewer_tH4H · 2024-11-04

**Soundness:** 2
**Presentation:** 3
**Contribution:** 2
**Rating:** 6
**Confidence:** 3

**Summary:**

Black-box LLMs, like GPT-4, are powerful tools that have shown promise in a wide range of tasks. However, it is difficult to control these models. Prior works have used in-context prompting to achieve this. In contrast, this paper trains a lightweight LLM, called Martryoshka, that controls the larger black box LLM to generate output that matches certain preferences. They provide an algorithm for training this lightweight LLM controller, and show the applicability of their algorithm in a wide range of tasks.

**Strengths:**

1. The paper is well-written and easy to follow.
2. The proposed method is simple and easy to follow.
3. Tested on mathematical reasoning, planning and personalization — 3 sufficiently different benchmarks, showing broad applicability of the paper’s methods.

**Weaknesses:**

**(Major concerns)**

Major concerns I have about this paper are:

1. The appendices seem to imply the method is much more complicated in practice to implement, than the main paper suggests. In fact, there are many key implementation steps without which the method probably doesn’t work. I would mention one:

> In cases where no positive generation exists for some data points, we utilize a more powerful model, such as gpt-4o, to produce several strong intermediate generations, thereby increasing the likelihood of obtaining positive samples.

The paper does not mention (or I could not find) what fraction of data is generated by gpt-4o/why is this needed at all. If data is generated by gpt-4o, then the paper’s method becomes distillation and not self-improvement, which is very different from what I first understood the paper to be.


2. The examples provided in the appendix raised more questions.

E.g., in page 26, for GSM-HARD, the following is the decomposition by Martryoshka:

Let’s break down this problem:

> 1. How much does the lemon tree cost to plant? 2. How much money will Carlos earn from selling lemons each year? 3. How much money will Carlos spend on watering and feeding the tree each year? 4. How many years will it take for Carlos to start earning money on the lemon tree?

The same done by the vanilla model is:

> Let’s break down this problem: 1. How much does the lemon tree cost to plant? 2. How much money will Carlos earn from selling lemons each year? 3. How much will it cost Carlos to water and feed the tree each year? 4. How many years will it take before Carlos starts earning money on the lemon tree?

The only difference between these two seems to be in slight differences in wording in question 3. Do the authors have any intuition on why such slight differences result in correct vs incorrect answers?


3. What is the data generation/API and training cost for these models?

4. Finally, does this method generalize? Eg given a Martryoshka model that has been trained to generate guidance on one task, generalize to an unseen task? If not, then every downstream user needs to train their own Martryoshka model every time they face a new task, which can be a weakness of this paper.

**(Minor concerns)**

>  Martryoshka adopts on-policy learning to iteratively enhance training data quality, inherently self-improving intermediate guidance for the continual enhancement of black-box LLM capabilities.

The paper mentions collecting on-policy data to improve training data quality. Citations for prior work that focuses on this issue is important. For example, [1] talks about online DPO being superior to the offline variant, [2] talks about the general importance of on-policy data for LLM alignment, [3] talks about the importance of online data from a theoretical perspective. Adding discussions on these, plus other relevant work would be important.

Also an interesting direction to think about for the authors is [5]. The authors are not training/updating the weights of the environment, but the idea of training an attacker or in this case controller policy, to drive the environment into behaving a certain way, has been studied in red-teaming/jail breaking literature and should be cited/credited as such. Similar ideas also appear in [6], though they do not train the controller policy as far as I understand.

**Questions:**

**(Question 1: choice of method)**
Any particular reason the authors employ iterative DPO, instead of directly using an online RL method like REINFORCE/PPO?

**(Question 2: Understanding multi-turn interaction in the paper)**

I do not understand the purpose of multi-turn interactions. Could the authors give an example trajectory obtained from the multi-turn interaction? I imagine the steps are:

1. the controller breaks the task into subgoals
2. sends it to the black-box LLM
3. obtains a response from the black box LLM, measure task success
4. The controller receives task success as the observation, and generates another break down of the task into subgoals?

Did I understand it correctly? If so, why is the controller policy only trains using single turn interactions in equation 9 and iterative guidance optimization? If I understand it correctly, the authors only train the model on positive guidance vs negative guidance, in a single turn setting, using DPO. This improves the controller at generating guidance at a single turn, but one can do more: if one uses the entire trajectory, define trajectory level reward, and use the multi-turn DPO formulation from Agent-Q [4], one can ideally create a controller policy that can keep improving the model’s generation using a multi-turn interaction system at test time, instead of just single step guidance generation. I am confused: could the authors clarify whether they indeed use a multi-turn DPO/multi-turn interaction at inference time (all the examples I saw on the appendix are of single turn interactions), or the multi-turn interaction is only used to generate more data? In that case, I do not see the purpose of using multi-turn interactions, once could attain as much data by just generating the single turn interaction more times.


# References

[1] Is DPO Superior to PPO for LLM Alignment? A Comprehensive Study, https://arxiv.org/abs/2404.10719

[2] Preference Fine-Tuning of LLMs Should Leverage Suboptimal, On-Policy Data, https://arxiv.org/abs/2404.14367

[3] The Importance of Online Data: Understanding Preference Fine-tuning via Coverage, https://arxiv.org/abs/2406.01462

[4] Agent Q: Advanced Reasoning and Learning for Autonomous AI Agents, https://arxiv.org/abs/2408.07199

[5] Red Teaming Game: A Game-Theoretic Framework for Red Teaming Language Models, https://arxiv.org/html/2310.00322v3

[6] Automated Black-box Prompt Engineering for Personalized Text-to-Image Generation, https://arxiv.org/abs/2403.19103

---

> ### Author Response · Authors · 2024-11-22
>
> We sincerely appreciate Reviewer tH4H for the insightful feedback. Your suggestions and insights are very helpful for further enhancing the submission quality. Please find the responses below:
>
> **[Major concerns 1]**
>
> Thank you for the suggestion. In practice, we only utilized the GPT-4o model in the DPO training set for the LaMP dataset. This was because the LaMP dataset lacks ground truth annotations, and in cases where the original black-box model could not generate sufficient positive data, we leveraged the GPT-4o model to produce additional positive samples needed for DPO training. Therefore, the use of the GPT-4o model was simply a design choice for data annotation.
>
> If alternative options were available, such as high-quality human-annotated data, ground truth chain-of-thoughts from GSM8K, or interaction-based feedback from Alfworld to determine the quality of data, we could entirely bypass the use of the GPT-4o model.
>
> In fact, even without the GPT-4o model, we can still achieve sufficiently or better positive results on the LaMP dataset. As shown in Table A, although slightly lower than the metrics reported using GPT-4o for LaMP-1 and LaMP-2, the results get even better on LaMP-3 and LaMP-4 and remain significantly better than other baselines on all LaMP tasks. As for the fraction, we observe that less than 5% data is from gpt-4o, further claiming that this setting is not needed. This demonstrates that our approach works well even without the assistance of a powerful model. In this case, where the powerful model is removed, it further highlights that the improvement in experimental performance stems from **self-improvement** rather than being directly distilled from a powerful model like GPT-4o.
>
> **Table A: Experimental results on the LaMP benchmark without GPT-4o-based labeling**
>
> |                                 | LaMP-1(ACC/F-1) | LaMP-2N(ACC/F-1) | LaMP-2M(ACC/F-1) | LaMP-3⬇️(MAE/RMSE) | LaMP-4(R-1/R-L/BLEU) |
> | ------------------------------- | ------------------- | -------------------- | -------------------- | --------------------- | ------------------------ |
> | LLaMA3-8B-Instruct              | 0.611/0.611         | 0.807/0.575          | 0.496/0.432          | 0.311/0.636           | 0.131/0.120/1.341        |
> | gpt-4o-mini                     | 0.624/0.624         | 0.775/0.559          | 0.496/0.443          | 0.316/0.645           | 0.143/0.130/1.968        |
> | gpt-4o-mini(zero-shot)          | 0.514/0.513         | 0.655/0.473          | 0.413/0.325          | 0.371/0.673           | 0.132/0.116/0.992        |
> | Martryoshka with 4o-Labeling    | **0.640/0.640**     | **0.832/0.614**      | **0.535/0.475**      | **0.282/0.588**       | **0.171/0.157/4.144**    |
> | Martryoshka without 4o-Labeling | **0.640/0.639**     | **0.823/0.607**      | **0.527/0.465**      | **0.277/0.581**       | **0.174/0.160/4.298**    |
>
> **[Major concerns 2]**
>
> We apologize for the confusion. On page 26 of the updated manuscript, we have included a new case study for the GSM_Hard task. In this case, Matryoshka showcases its ability to reflect on issues in the previous decomposition and provide a more precise re-decomposition, ultimately guiding the black-box LLM to successfully solve the problem.
>
> This new case highlights two key points:
>
> 1. Using a white-box LLM to drive a black-box LLM is a highly promising direction, significantly outperforming CoT and PaL methods that rely solely on the intrinsic capabilities of the black-box LLM.
> 2. The white-box LLM's ability to learn within this framework is equally promising. Through learning, Matryoshka acquires reflective capabilities and the ability to revise previous decompositions—skills that the vanilla LLaMA3-8B-Instruct lacks—further enhancing its ability to guide the black-box LLM in problem-solving.
>
> Besides, we have also updated the white-box LLM multi-turn interaction prompts for both the AlfWorld and GSM tasks in Appendix G.

---

> ### Author Response · Authors · 2024-11-22
>
> **[Major concerns 3]**
>
> The data generation cost was calculated by aggregating the total token consumption statistics provided by Azure API and subsequently applying the cost per token (gpt-3.5-turbo-0125, gpt-4o-mini) as specified in the OpenAI official documentation. The cost for processing 1M tokens, as detailed in Table B, served as the basis for this calculation.
>
> **Table B: Detailed API cost per 1 million tokens [1]**
>
> | Model         | Input cost ($) / 1M tokens | Output cost ($) / 1M tokens |
> | ------------- | -------------------------- | --------------------------- |
> | gpt-3.5-turbo | 3.0                        | 6.0                         |
> | gpt-4o-mini   | 0.15                       | 0.6                         |
>
> For the AlfWorld dataset, the entire training set consists of 8,808 samples. On average, using GPT-3.5-turbo to sample 100 examples costs approximately **$3.20**, making the estimated cost for complete data collection **$282**.
>
> For the GSM8K dataset, the full training set comprises 7,473 samples. The average cost for sampling 100 examples using GPT-3.5-turbo is **$1.215**, resulting in an estimated total cost for data collection of **$90.80**. *In comparison, fine-tuning the gpt-3.5-turbo with the Azure OpenAI service costs $216.50, and requires hourly payment in deployment for inference.*
>
> For LaMP-1, LaMP-2M, LaMP-2N, LaMP-3, LaMP-4, we use gpt-4o-mini for data generation. The total costs are separately **\$6.144,  1.882, 2.348, 8.111, 10.022**, with 5252, 2719, 2369, 8506, 12518 generated data samples.
>
> During the training phase, we used four H100 GPUs for two rounds of DPO training. The process took approximately 1.5 hours for AlfWorld and GSM8K, resulting in a total training cost of **6 GPU hours**. It took separately 4 gpu h, 4 gpu h, 4 gpu h, 8 gpu h, 12 gpu h for the training process of LaMP-1, -2M, -2N, -3, -4.
>
> Reference:
>
> [1] https://openai.com/api/pricing/
>
> **[Major concerns 4]**
>
> We are grateful for the reviewer's insightful suggestion. We envision our model as a universal white-box assistant capable of being seamlessly integrated into new tasks in a plug-and-play manner while effectively guiding large models. We are currently exploring strategies to achieve this goal and will continue to pursue lightweight, versatile control capabilities for smaller models in future research.
>
> **[Minor concerns]**
>
> We first thank the reviewer for the great insights. The citation of relevant papers on collecting on-policy data is indeed important, and we will add the relevant citations to our revised version.
>
> Specifically, [1] mentions that online DPO performs better than offline DPO. The insight/observation is compatible with our work, which utilizes online DPO for multiple iterations for continual enhancement of black-box LLM capabilities. Also worth mentioning, our method can accommodate different design choices for DPO, SimPO and so on, while our key focus is the proposed “LLM drives LLM” controller.
>
> [2] and [3] both stress the importance of on-policy/online data, from practical and theoretical perspectives. We will add discussions on these relevant work in our revised version.
>
> [5] and [6] offer interesting insights that there are various ways of training and controlling with regard to what to train and potential applications.
>
> Thanks again for your insightful suggestions, and we will add these meaningful and highly relevant discussions and citations into our paper.
>
> Reference:
>
> [1] Is DPO Superior to PPO for LLM Alignment? A Comprehensive Study, https://arxiv.org/abs/2404.10719
>
> [2] Preference Fine-Tuning of LLMs Should Leverage Suboptimal, On-Policy Data, https://arxiv.org/abs/2404.14367
>
> [3] The Importance of Online Data: Understanding Preference Fine-tuning via Coverage, https://arxiv.org/abs/2406.01462
>
> [4] Agent Q: Advanced Reasoning and Learning for Autonomous AI Agents, https://arxiv.org/abs/2408.07199
>
> [5] Red Teaming Game: A Game-Theoretic Framework for Red Teaming Language Models, https://arxiv.org/html/2310.00322v3
>
> [6] Automated Black-box Prompt Engineering for Personalized Text-to-Image Generation, https://arxiv.org/abs/2403.19103

---

> > ### Author Response · Authors · 2024-11-22
> >
> > **[Question 1]**
> >
> > We appreciate the suggestion and would like to elaborate on the reinforcement learning method we chose. First, we improved upon the basic DPO method by introducing an iterative guidance optimization approach that interleaves data sampling with model training steps. Through extensive experiments, we demonstrated that this method—which incorporates multi-turn interactions with the environment—significantly outperforms simple DPO training.
> >
> > Furthermore, it is important to emphasize that the core contribution of our work lies in adapting the pioneering controller-generator framework. This framework is highly flexible and enables controllable multi-turn generation, significantly enhancing the ability of black-box large language models to handle complex, long-horizon tasks.
> >
> > The specific RL method used to train the controller model is simply a design choice and not the primary focus of our work. Nevertheless, to showcase the adaptability of our framework, we conducted experiments on all three datasets using SimPo [1] as the RL method. The results, presented below, demonstrate that our framework consistently delivers strong performance across different RL methods, further validating its effectiveness and generality.
> >
> > **AlfWorld**
> >
> > | Methods(↓) Tasks(→)       | Pick       | Clean     | Heat       | Cool       | Examine   | Pick Two  | All       |
> > | ------------------------- | ---------- | --------- | ---------- | ---------- | --------- | --------- | --------- |
> > | Adaplanner                | 100.00     | 93.55     | 78.26      | 95.24      | 66.67     | 88.24     | 88.06     |
> > | Matry+simpo               | **100.00** | **93.55** | **95.65**  | **95.24**  | **77.78** | **88.24** | **92.54** |
> > | w/o $1^{st}$-round SimPO  | 100.00     | 93.55     | 86.96      | 95.24      | 55.56     | 88.24     | 88.06     |
> > | w/o Guidance Optimization | 100.00     | 93.55     | 91.30      | 85.71      | 11.11     | 88.24     | 81.34     |
> > | Matry + dpo               | **100.00** | **93.55** | **100.00** | **100.00** | **83.33** | **88.24** | **94.78** |
> >
> >
> >
> > **GSM**
> >
> > | Dataset(→) Methods(↓) | GSM8K(gpt-3.5) | GSM8K(4o-mini) | GSM-HARD(gpt-3.5) | GSM-HARD(4o-mini) |
> > | --------------------- | -------------- | -------------- | ----------------- | ----------------- |
> > | CoT                   | 0.809          | 0.932          | 0.406             | 0.5               |
> > | Least-to-Most         | 0.811          | 0.908          | 0.425             | 0.498             |
> > | PAL                   | 0.802          | 0.920          | 0.638             | 0.748             |
> > | PAL_{Self-Debug}      | 0.864          | 0.943          | 0.701             | 0.774             |
> > | Matry+simpo           | **0.908**      | **0.950**      | **0.731**         | **0.789**         |
> > | Matry+dpo             | **0.911**      | **0.956**      | **0.738**         | **0.779**         |
> >
> >
> >
> > **LaMP**
> >
> > |                 | Acc/F-1 ⬆️       | Acc/F-1 ⬆️       | Acc/F-1 ⬆️       | mae/rmse ⬇️      | R-1/R-L/BLEU ⬆️        |
> > | --------------- | --------------- | --------------- | --------------- | --------------- | --------------------- |
> > |                 | LaMP1           | LaMP-2N         | LaMP-2M         | LaMP-3          | LaMP-4 prompt1        |
> > | llama3 + 4omini | 0.611/0.611     | 0.807/0.575     | 0.496/0.432     | 0.311/0.636     | 0.131/0.120/1.341     |
> > | 4omini + 4omini | 0.624/0.624     | 0.775/0.559     | 0.496/0.443     | 0.316/0.645     | 0.143/0.130/1.968     |
> > | Matry+dpo   | **0.640/0.640** | **0.832/0.614** | **0.535/0.475** | **0.282/0.588** | **0.171/0.157/4.144**     |
> > | Matry+simpo | **0.628/0.628**     | **0.826/0.598**     | **0.522/0.461**     | **0.294/0.614**     | **0.180/0.167/4.997** |
> >
> > Reference:
> >
> > [1] Meng, Y., Xia, M. and Chen, D., 2024. Simpo: Simple preference optimization with a reference-free reward. *arXiv preprint arXiv:2405.14734*.

---

> > > ### Author Response · Authors · 2024-11-22
> > >
> > > **[Question 2]**
> > >
> > > We appreciate the reviewer’s insightful observation regarding the controller’s multi-turn interactions. The original goal of our project was to design the controller with multi-turn interaction capabilities. However, at the time of submission, our main experiments were mainly conducted using single-turn data to train the controller. During the inference stage, the controller model also performed a single decomposition of the problem. While this approach already demonstrated strong performance on reasoning, planning, and personalization tasks, there remains significant room for improvement.
> > >
> > > Currently, we have incorporated multi-turn data collection into our pipeline. Specifically, if the white-box LLM’s initial decomposition fails to guide the black-box LLM to solve the problem, the white-box LLM is prompted to re-decompose the question based on the previous trajectory. We then combine the context of the problem, the trajectory from both attempts into a single multi-turn data.
> > >
> > > During inference, we have also integrated multi-turn interactions. If the subgoals provided by the controller fail to guide the black-box model to solve the problem, the controller model reflects on its errors and re-decomposes the problem into subgoals to guide the black-box model.
> > >
> > > By introducing multi-turn interactions into both the training and inference stages of the controller model, we achieved improved performance on reasoning and planning tasks. The results are summarized in the table below:
> > >
> > > **AlfWorld**
> > >
> > > | **Methods(↓) Tasks(→)** | **Pick**   | **Clean** | **Heat**   | **Cool**  | **Examine** | **Pick Two** | **All**   |
> > > | ----------------------- | ---------- | --------- | ---------- | --------- | ----------- | ------------ | --------- |
> > > | Single-Turn             | 100.00     | 93.55     | 100.00     | 90.48     | 100.00      | 88.24        | 95.52     |
> > > | **Multi-Turn(Turn=2)**  | **100.00** | **93.55** | **100.00** | **95.24** | **100.00**  | **88.24**    | **96.27** |
> > >
> > > **GSM**
> > >
> > > | **Dataset(→) Methods(↓)** | **GSM8K(gpt-3.5)** | **GSM8K(4o-mini)** | **GSM-HARD(gpt-3.5)** | **GSM-HARD(4o-mini)** |
> > > | ------------------------- | ------------------ | ------------------ | --------------------- | --------------------- |
> > > | Single-Turn               | 0.911              | 0.956              | 0.738                 | 0.779                 |
> > > | **Multi-Turn(Turn=2)**    | **0.931**          | **0.964**          | **0.761**             | **0.801**             |
> > >
> > > As seen, the addition of Multi-Turn interactions has resulted in enhanced performance across all subtasks compared with Single-Turn interactions. In the updated version of the manuscript, we have added prompts for the controller model's multi-turn interactions on the GSM-Hard and ALFWorld datasets in Appendix G.

---

> > > > ### Comment · Reviewer_tH4H · 2024-11-25
> > > >
> > > > I thank the authors for their thoughtful rebuttal. Based on their answer, I have increased my score from 5 to 6.

---

> ### Author Response · Authors · 2024-11-25
>
> Dear Reviewer tH4H,
>
> Thank you very much for taking the time to review our rebuttal and offering insightful feedback. Also, thank you for raising the score!
>
> Please don't hesitate if you have any further questions.
>
> Best regards,
>
> Authors

---

### Official Review · Reviewer_22gq · 2024-11-04

**Soundness:** 2
**Presentation:** 2
**Contribution:** 1
**Rating:** 3
**Confidence:** 5

**Summary:**

This paper presents an approach to enhance the capabilities of black-box Large Language Models (LLMs) by employing a lightweight white-box LLM as a controller. The proposed framework, Martryoshka, addresses the opacity of black-box LLMs by decomposing complex tasks into a series of intermediate outputs, which are used to guide the black-box LLM through iterative interactions. This controller-generator framework allows for controllable multi-turn generation and self-improvement in optimizing intermediate guidance, leading to improved performance in reasoning, planning, and personalization tasks.

**Strengths:**

The method proposed in this paper is successfully applied in a white-box + black box framework, and  shows its  effectiveness in reasoning, planning, and personalization tasks compared with black-box LLMs.

**Weaknesses:**

1. The key idea of Martryoshka is to train a white-box LLM to generate CoT or planning steps, then combine them with the original prompt and send them to black-box LLMs. The  methodology used to train white-box LLMs is a SFT phase combined with a DPO phase, which is a common and  trivial way for enhence domain specific  abilities (like math, coding, instruction following etc.) of certain LLMs. The method used in Martryoshka for training white-box LLMs simply changes the enheced abilities to 'CoT generation' or 'planing generation'. There is concern about the novelty of this methodology.
2. The proposed idea of control black-box LLMs with white-box is not convincing in practical. A potential  problem is if we already have SFT data and pair data for RLHF of a certain task, why don't we simply  train a domain-specify model by using 'while-box' LLMs? In many benchmarks, 'while-box' LLMs are already superior than black-box LLMs like GPT-4o in specific domains. For example, Llama3.1  series outperform GPT-4o in 10/13 benchmarks, including the GSM8K which is tested in this paper.

**Questions:**

I  suggest the author discuss the following questions：
1. What specific advantages does your white-box + black-box framework offer over fine-tuning a white-box model?
2. Are there scenarios where your approach would be preferable to domain-specific fine-tuning of white-box models?
3. How does your method compare to fine-tuned white-box models on the same tasks?

---

> ### Author Response · Authors · 2024-11-22
>
> Thank you for your detailed suggestions and comments. Initially, we would like to clarify the focus of this paper on driving black-box LLMs to obtain desired capabilities following a series of existing works [1][2][3]. Currently, SOTA black-box LLMs still show better performances than open-source LLMs on different tasks, including all three tasks we use in the paper [4][5][6], which motivates us to find a valid method to drive the generation of black-box LLMs to achieve better performance on specific tasks. By leveraging a lightweight white-box LLM to control the black-box LLM, we can demonstrate the black-box LLM's broad applicability and exceptional performance across various tasks. Additionally, by training the white-box LLM to drive black-box LLM, we enable personalized utilization of the black-box LLM to better suit specific needs. Experiments also show that with the same budgets in model parameters and training data, our white-box LLM + black-box LLM framework outperforms simply fine-tuning the white-box LLM. Please find the detailed corresponding responses below:
>
> [1] Huang, Yangsibo, Daogao Liu, Zexuan Zhong, Weijia Shi, and Yin Tat Lee. "$ k $ NN-Adapter: Efficient Domain Adaptation for Black-Box Language Models." arXiv preprint arXiv:2302.10879 (2023).
>
> [2] Sun, Haotian, Yuchen Zhuang, Wei Wei, Chao Zhang, and Bo Dai. "BBox-Adapter: Lightweight Adapting for Black-Box Large Language Models." In Forty-first International Conference on Machine Learning.
>
> [3] Zhuang, Yuchen, Haotian Sun, Yue Yu, Rushi Qiang, Qifan Wang, Chao Zhang, and Bo Dai. "Hydra: Model factorization framework for black-box llm personalization." arXiv preprint arXiv:2406.02888 (2024).
>
> [4] https://paperswithcode.com/sota/arithmetic-reasoning-on-gsm8k
>
> [5] https://paperswithcode.com/dataset/alfworld
>
> [6] Salemi, Alireza, Sheshera Mysore, Michael Bendersky, and Hamed Zamani. "Lamp: When large language models meet personalization." arXiv preprint arXiv:2304.11406 (2023).
>
> **[Weakness 1]**
>
> Thank you for your valuable feedback. We would like to emphasize that the primary contribution of our work lies in treating black-box LLMs as environments and training a lightweight white-box LLM controller to guide the outputs of black-box LLMs in alignment with specific preferences. This enables controllable multi-turn generation and enhances the capability of black-box LLMs in handling complex, long-horizon tasks. Currently, state-of-the-art black-box LLMs outperform open-source LLMs across various tasks, such as Code Generation on HumanEval [1]. This motivates us to develop an effective approach to better control the generation of black-box LLMs for specific domains or tasks.
>
> From a generalization perspective, existing works, such as Adaplanner[2], Hydra[3], focus on limited use cases. In contrast, our method has proven effective across a broader range of tasks, including reasoning, planning, and personalization—covering a scope that no prior work has comprehensively addressed. Moreover, compared to previous approaches, our proposed method requires a lighter-weight white-box controller and a smaller amount of data.
>
> Additionally, our approach is highly flexible, allowing it to be applied to different black-box or open-source LLMs in a plug-and-play manner. Finally, within our framework, the intermediate responses generated by the white-box LLMs can take various forms beyond CoT generation and planning generation. We appreciate the reviewer's insightful questions and will explore additional potential representations in future research.
>
> Reference:
>
> [1] https://paperswithcode.com/sota/code-generation-on-humaneval
>
> [2] Sun, Haotian, Yuchen Zhuang, Lingkai Kong, Bo Dai, and Chao Zhang. "Adaplanner: Adaptive planning from feedback with language models." *Advances in Neural Information Processing Systems* 36 (2024).
>
> [3] Zhuang, Yuchen, Haotian Sun, Yue Yu, Rushi Qiang, Qifan Wang, Chao Zhang, and Bo Dai. "Hydra: Model factorization framework for black-box llm personalization." arXiv preprint arXiv:2406.02888 (2024).

---

> > ### Author Response · Authors · 2024-11-22
> >
> > **[Weakness 2]**
> >
> > We appreciate your insightful question regarding the direct training of a domain-specific white-box LLM. A key advantage of our controller-generator framework lies in its ability to ensure both broad applicability and outstanding performance across various tasks. Since the black-box LLM already possesses strong baseline capabilities in diverse downstream tasks, we only need to train a lightweight controller model with a small amount of data and cost-effective computational resources to effectively harness and drive the intrinsic abilities of the black-box LLM.
> >
> > Additionally, the term "black-box LLMs" here generally refers to models that are difficult to fine-tune, such as LLaMA3.1-405B-Instruct or LLaMA3.1-70B-Instruct, which require extremely costly computational resources. This further highlights the practical importance of our approach.
> >
> > Moreover, black-box LLMs continue to excel across a wide range of tasks. For instance, on GSM8K, only a fine-tuned 70B LLaMA3.1 model achieves performance comparable to GPT-4 [1], and the fine-tuning process itself demands significant computational resources. This motivates us to develop an effective method that leverages a smaller white-box LLM to guide and enhance the performance of existing black-box LLMs on various tasks.
> >
> > Reference:
> >
> > [1]  https://paperswithcode.com/sota/arithmetic-reasoning-on-gsm8k
> >
> > **[Questions 1 & 2]**
> >
> > We appreciate the suggestion and would like to outline a real-world scenario where our white-box + black-box framework is preferable to directly fine-tuning a white-box model, along with its specific advantages.
> >
> > First, our approach enables the white-box model to remain lightweight while fully leveraging the capabilities of the black-box model. For example, in Apple's cloud system services, simpler tasks like local photo editing and management can be handled directly by the device's on-board capabilities. In contrast, more complex tasks, such as database-supported facial recognition, rely on large-scale cloud servers. Here, the role of the large servers parallels that of the black-box LLM, while the user’s local device resembles the more lightweight white-box LLM. By combining the strengths of a powerful black-box model with a lightweight white-box model tailored to specific user needs, the framework achieves better overall performance.
> >
> > Furthermore, fine-tuning a white-box LLM will lose the generalization capabilities of multi-turn interaction to sample higher-quality data, which is also a key component of our work.
> >
> > **[Questions 3]**
> >
> > Thank you for the suggestion. We have included additional evaluations of the LLaMA3-8B-Instruct model on the GSM8K and GSM-Hard datasets. Following the same experimental settings as outlined in the paper, we employed a three-shot prompt design and incorporated closed-loop refinement via PAL during the inference stage.
> >
> > We evaluated the performance of both the vanilla LLaMA3-8B-Instruct model and the DPO fine-tuned LLaMA3-8B-Instruct model as well as SFT fine-tuned LLaMA3-8B-Instruct model. As shown in Table A, even when utilizing a relatively less powerful black-box model such as GPT-3.5, our white-box + black-box framework outperforms the powerful SFT fine-tuned white-box model by **6.4%** on the GSM8K dataset and **6.3%** on the GSM-Hard dataset. These results strongly demonstrate that our black-box + white-box approach is significantly more effective than directly fine-tuning the white-box model.
> >
> > **Table A: Comparison with fine-tuned white-box models on GSM task**
> >
> > |                                             | **GSM8K** | **GSM-Hard** |
> > | ------------------------------------------- | --------- | ------------ |
> > | LLaMA3-8B-Instruct                          | 0.841     | 0.661        |
> > | SFT fine-tuned LLaMA3-8B-Instruct           | 0.847     | 0.675        |
> > | DPO fine-tuned LLaMA3-8B-Instruct           | 0.832     | 0.641        |
> > | Matryoshka(LLaMA3-8B-Instruct) + gpt3.5     | **0.911** | **0.738**    |
> > | Matryoshka(LLaMA3-8B-Instruct) + gpt4o-mini | **0.956** | **0.779**    |

---

> > > ### Author Response · Authors · 2024-11-25
> > > **Gentle reminder: The interactive discussion period will end in less than two days**
> > >
> > > Dear Reviewer 22gq,
> > >
> > > Thank you again for your time and efforts in reviewing our paper.
> > >
> > > As the discussion period draws close, we kindly remind you that two days remain for further comments or questions. We would appreciate the opportunity to address any additional concerns you may have before the discussion phase ends.
> > >
> > > Thank you very much.
> > >
> > > Best regards,
> > >
> > > Authors

---

> > ### Comment · Reviewer_22gq · 2024-11-28
> >
> > I've carefully read the authors' responses, but I still have concerns about the effectiveness of this training method. If we want to guide black-box LLMs to generate better responses, there are hundreds of prompting methods available. When incorporating training into our process, there are numerous methods for enhancing domain-specific abilities with white-box LLMs, albeit with risks of data leakage and API costs for black-box LLMs. Therefore, I will maintain my score at 3 for the reasons stated above.

---

> > > ### Author Response · Authors · 2024-11-28
> > >
> > > We appreciate Reviewer 22gq’s response to our work. However, we respectfully disagree, and our reasons are as follows:
> > >    1. The reviewer suggests that black-box LLMs can be guided to generate better responses using prompting techniques. However, we want to emphasize that our approach is not merely about crafting specific prompts but involves training a **sequential controller**, which is fundamentally different. For example, our pipeline incorporates multi-turn interactions between the controller and the black-box LLM. While prompts can enhance an LLM's instruction-following ability, our method introduces significantly more **flexibility** by enabling the black-box LLM to interact with its environment. This added control translates into **better performance** across various tasks, as demonstrated in our experiments.
> > >    2. **Regarding improving black-box LLM performance without training,** existing methods primarily rely on in-context learning and carefully designed instructions. Compared to previous state-of-the-art results:
> > >       - In **Planning tasks,** our method outperforms Adaplanner [1] by 7.5 percentage points, achieving better performance across all subtasks (see Table 4).
> > >       - In **Reasoning tasks,** we surpass PAL [2] and CoT [3] by 5 and 10 percentage points, respectively, when using gpt-3.5-turbo and gpt-4o-mini as the black-box LLM (see Table 3).
> > >       - In **Personalization tasks,** we achieve significant improvements over RAG [4] and PAG [5], with consistent gains across LaMP-1 to LaMP-4 datasets.
> > >    3. **For training domain-specific white-box LLMs,** fine-tuning methods typically fall into two categories:
> > >       - (a) Supervised learning, such as SFT.
> > >       - (b) Reinforcement learning with human feedback (RLHF), such as DPO or SIMPO.
> > >
> > >         Compared to both (a) and (b), our method delivers superior results in the **GSM task**, surpassing them by over 6 percentage points, as shown in **Table A**. Moreover, the concern of data leakage also exists for white-box models.
> > >    4. **Regarding cost,** the API usage cost is manageable. We provide data collection costs for all three tasks:
> > >       - For LaMP, the total cost can be kept **under $10** by using the most affordable gpt-4o-mini API.
> > >       - For Alfworld and GSM tasks, even with the slightly more expensive gpt-3.5-turbo API, the total cost remains **around $100**.
> > >    5. **If there are any other methods not covered in points 2 and 3, please let us know, and we are more than willing to add additional experiments for comparison.**
> > >
> > > Reference:
> > >
> > > [1] Sun, Haotian, Yuchen Zhuang, Lingkai Kong, Bo Dai, and Chao Zhang. "Adaplanner: Adaptive planning from feedback with language models." *Advances in Neural Information Processing Systems* 36 (2024).
> > >
> > > [2] Gao, Luyu, Aman Madaan, Shuyan Zhou, Uri Alon, Pengfei Liu, Yiming Yang, Jamie Callan, and Graham Neubig. "Pal: Program-aided language models." In *International Conference on Machine Learning*, pp. 10764-10799. PMLR, 2023.
> > >
> > > [3] Wei, Jason, Xuezhi Wang, Dale Schuurmans, Maarten Bosma, Fei Xia, Ed Chi, Quoc V. Le, and Denny Zhou. "Chain-of-thought prompting elicits reasoning in large language models." *Advances in neural information processing systems* 35 (2022): 24824-24837.
> > >
> > > [4] Salemi, Alireza, Sheshera Mysore, Michael Bendersky, and Hamed Zamani. "Lamp: When large language models meet personalization." *arXiv preprint arXiv:2304.11406* (2023).
> > >
> > > [5] Richardson, Chris, Yao Zhang, Kellen Gillespie, Sudipta Kar, Arshdeep Singh, Zeynab Raeesy, Omar Zia Khan, and Abhinav Sethy. "Integrating summarization and retrieval for enhanced personalization via large language models." *arXiv preprint arXiv:2310.20081* (2023).

---

> ### Author Response · Authors · 2024-11-25
>
> Hi Reviewer 22gq,
>
> We have carefully considered your valuable feedback and have submitted a detailed response addressing the points raised in your reviews. We believe our response clarifies several aspects of the paper, highlights its contributions, and our additional work (attached above and in the updated paper) addresses your concerns. It would be great if you could take some time to review our responses and let us know your feedback.
>
> Thanks in advance,
>
> Authors of paper **Martryoshka: Learning to Drive Black-Box LLMs with LLMs**

---

### Author Response · Authors · 2024-11-22

Dear reviewers,

We sincerely appreciate the time and effort dedicated to evaluating our work. We have summarized the additional experiments and analyses during the rebuttal phase, and we are committed to incorporating them in the revised manuscript.

Our newly added main experiments and analysis include:

- **Comparison with fine-tuned white-box models:**  We demonstrate that even when using a relatively less powerful black-box model, such as GPT-3.5, we can still outperform a fine-tuned white-box model of the same size by more than 6% on the GSM8K dataset.
- **Experiments on LaMP benchmark without gpt-4o-based labeling:** We show that even without the GPT-4o model, we can still achieve sufficiently or better positive results on the LaMP dataset and remain significantly better than other baselines on all LaMP tasks.
- **Experiments with different RL methods:** We implement another RL method, SimPo, within our framework, and it demonstrates significant effectiveness across all tasks. The results highlight the compatibility of our framework with various RL methods.
- **Multi-turn interactions for the white-box controller model:** We incorporate multi-turn interactions into both the data collection and inference stages. Experimental results show that this approach further improves performance on reasoning (by 2%) and planning (by 0.7%) tasks compared to single-turn interactions.
- **Comparison with directly using two black-box LLMs:** We summarize the results of directly using two black-box LLMs, including GPT-3.5 and the more powerful GPT-4o-mini, across various tasks. Our method consistently outperforms them on all tasks. We attribute this to the inability of black-box LLMs to interact effectively with the environment, which prevents them from performing efficient problem decomposition.

We would like to further emphasize our main contributions as follows:

- We propose Matryoshka, **one of the first modular frameworks** that employ a lightweight white-box LLM to drive the generation of a large-scale black-box LLM for complex problem solving;
- Matryoshka intuitively formulates **the white-box LLM as a controller** and **the black-box LLM as a component of the environment**, facilitating long-horizon controllable generation with environmental feedback;
- Matryoshka significantly **outperforms existing baselines** across three complex tasks including reasoning, planning, and personalization, introducing one of the first learning-based solutions that not only enhances the capabilities of black-box LLMs without requiring access to model parameters, but also facilitates online feedback with environmental interactions.



Please find the point-to-point response with additional details in the following rebuttal section. We sincerely hope that our responses have enhanced the paper's quality and addressed your concerns. If you have any additional suggestions or comments, please don't hesitate to share them. We look forward to engaging in a constructive discussion during the rebuttal phase. Thank you again for your understanding and consideration.

Best regards,

Authors

---

### Meta-Review · Area_Chair_iSmv · 2024-12-23

**Metareview:**

The paper proposes a method to control black-box LLMs. The core idea is to train a lightweight LLM (Martryoshka), that controls the larger black box LLM to generate output that matches certain preferences. Empirical evaluations are performed on three tasks and highlight potential of the framework.

The criticism of the paper mostly focuses on the novelty, including a couple of reviewers questioning the white label methodology. Also there are questions about the empirical evaluation.

My recommendation to reject this paper is primarily based on the exposition and clarity of the contribution. Most of the reviewers raise questions about the methodology, implying similarity to existing methods and asking for simpler / clearer description. My recommendation to the authors would be to clearly express the novel contributions.

**Additional Comments On Reviewer Discussion:**

While the authors provide a rich rebuttal, only two of the reviewers marginally increase their scores from 5 to 6, while the most critical reviewer maintained their score (3). One of the reviewers mention that he raised the score due to better baselines. I feel that the exposition can still be improved, hence my recommendation.

---

### Decision · Program_Chairs · 2025-01-22

Reject